# Cross-modal Prompts: Adapting Large Pre-trained Models for Audio-Visual Downstream Tasks

**Haoyi Duan**[1*]   **Yan Xia**[1*]   **Mingze Zhou**[1]   **Li Tang**[1]   **Jieming Zhu**[3]   **Zhou Zhao**[1,2†]

[1] Zhejiang University    [2]Shanghai Artificial Intelligence Laboratory    [3]Huawei Noah's Ark Lab

{haoyiduan075, xiayan.zju}@gmail.com

{3200102572, tanglzju, zhaozhou}@zju.edu.cn   jiemingzhu@ieee.org

## Abstract

In recent years, the deployment of large-scale pre-trained models in audio-visual downstream tasks has yielded remarkable outcomes. However, these models, primarily trained on single-modality unconstrained datasets, still encounter challenges in feature extraction for multi-modal tasks, leading to suboptimal performance. This limitation arises due to the introduction of irrelevant modality-specific information during encoding, which adversely affects the performance of downstream tasks. To address this challenge, this paper proposes a novel Dual-Guided Spatial-Channel-Temporal (DG-SCT) attention mechanism. This mechanism leverages audio and visual modalities as soft prompts to dynamically adjust the parameters of pre-trained models based on the current multi-modal input features. Specifically, the DG-SCT module incorporates trainable cross-modal interaction layers into pre-trained audio-visual encoders, allowing adaptive extraction of crucial information from the current modality across spatial, channel, and temporal dimensions, while preserving the frozen parameters of large-scale pre-trained models. Experimental evaluations demonstrate that our proposed model achieves state-of-the-art results across multiple downstream tasks, including AVE, AVVP, AVS, and AVQA. Furthermore, our model exhibits promising performance in challenging few-shot and zero-shot scenarios. The source code and pre-trained models are available at https://github.com/haoyi-duan/DG-SCT.

## 1   Introduction

With the increasing availability of hardware resources, large-scale models [20, 4, 3] pre-trained on extensive data have achieved significant advancements in various multi-modal tasks [26, 5]. Nonetheless, since these models are primarily pre-trained on single modality, they may not be optimally suited for current multi-modal downstream tasks [33, 32, 43, 14]. As depicted in Fig 1 (a), the pre-trained model equally extracts visual features and directly passes them to downstream tasks. However, when perceiving the roaring sound of an engine, the visual region depicting a "car" should receive more attention than the region of "trees". Simultaneously, when observing the car, it is crucial to concentrate on the audio segments of the engine sound. Therefore, the encoder should not only equally extract modal-specific information from the current modality, but also highlight information related to other modalities to enhance feature fusion across diverse modalities in downstream tasks. Retraining these large models based on downstream tasks would impose an unaffordable burden [41, 22, 39], leading recent works to explore methods for fine-tuning pre-trained models on downstream tasks without full retraining, showing promising progress. However, these CLIP-based methods have primarily focused on text-image tasks [46, 45, 7, 12] , while overlooking another important

---

*Equal contribution.

†Corresponding author.

37th Conference on Neural Information Processing Systems (NeurIPS 2023).

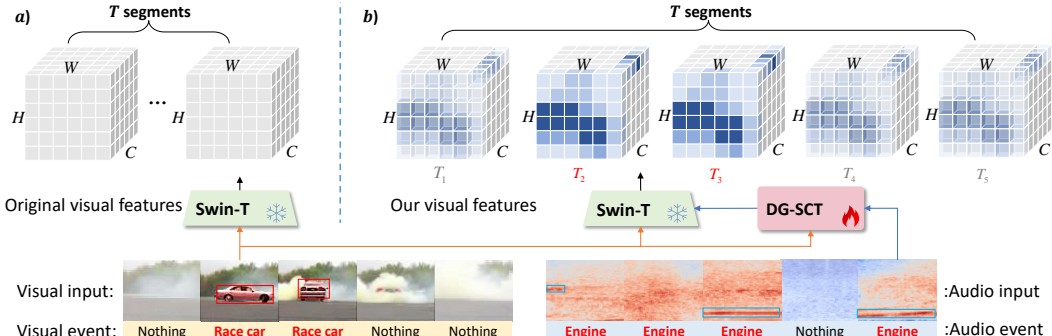

Figure 1: Our **Spatial** and **Temporal** attention can focus on important regions and moments in video and emphasize critical timestamps and frequencies in audio; **Channel** attention enhances the representations of audio and visual features. Take visual modality, our visual features contain fine-grained, task-specific information under the guidance of audio prompts.

multi-modal scenario: audio-visual tasks. Hence, in this paper, we primarily investigate how to utilize existing large-scale models, such as CLIP [26] and Swin-Transformer [20], to adaptively adjust the encoding features with the guidance of counterpart modality when encoding audio or visual information.

The success of prompt learning in large language models (LLMs) [13, 18, 19, 30, 21] has recently sparked growing research interest in multi-modal prompt learning, as seen in works such as CoOp [46], CoCoOp [45], CLIP-Adapter [7], DenseCLIP [27], and MaPLe [12]. While these approaches enable adaptive adjustment of input features using text-based prompt templates for downstream tasks, an important question arises: *Can audio or video serve as innovative prompt templates to enhance task comprehension for pre-trained models and guide adaptive feature extraction of the counterpart modality?* Our findings suggest a positive answer to this question.

In this paper, we present the **Dual-Guided Spatial-Channel-Temporal** (DG-SCT) attention mechanism, designed to adaptively adjust feature extraction of pre-trained models based on audio-visual input. Our work is motivated by a recent work, LAVisH [17], which introduces trainable layers with shared parameters in pre-trained models to enhance fusion of audio-visual features, demonstrating promising performance on various audio-visual downstream tasks with minimal additional parameters. However, LAVisH has a few limitations. First, it relies solely on a visual encoder to encode audio, which we argue is insufficient for capturing key audio features [10, 23]. Second, it only employs cross-attention in trainable layers to introduce information from different modalities, without explicitly highlighting crucial information within the current modality. By contrast, our approach incorporates cross-modal interaction layers into audio (HTS-AT [3]) and visual (ViT [4] or Swin-T [20]) pre-trained models, leveraging different modalities as prompts to focus on special aspects of the input features that are more relevant to the counterpart modal semantics across **spatial**, **channel**, and **temporal** dimensions. We term our proposed approach as "prompts" to denote the guidance provided to the trainable weights in preceding layers. It emphasizes utilizing audio and video cues to guide the representation of the counterpart modalities.

As depicted in Fig. 1, unlike previous audio and visual encoders, which generate audio and visual features separately and uniformly (Fig. 1 (a)), our features contain fine-grained, task-specific information at multiple levels by leveraging the guiding characteristics of multi-modal information [37, 36]. This enables efficient implementation of a wide range of downstream tasks. Notably, unlike previous CLIP works [46, 45] that offer unidirectional prompts, our approach introduces bidirectional prompts. This means that visual and audio modalities can mutually guide each other, facilitating enhanced feature extraction from the respective modalities.

In summary, this paper makes the following contributions:

- We highlight the limitations faced by large-scale pre-trained models in audio-visual downstream tasks, which hinder their optimal performance. To overcome these, we propose to utilize audio-visual features as novel prompts to fully leverage the feature extraction capa-

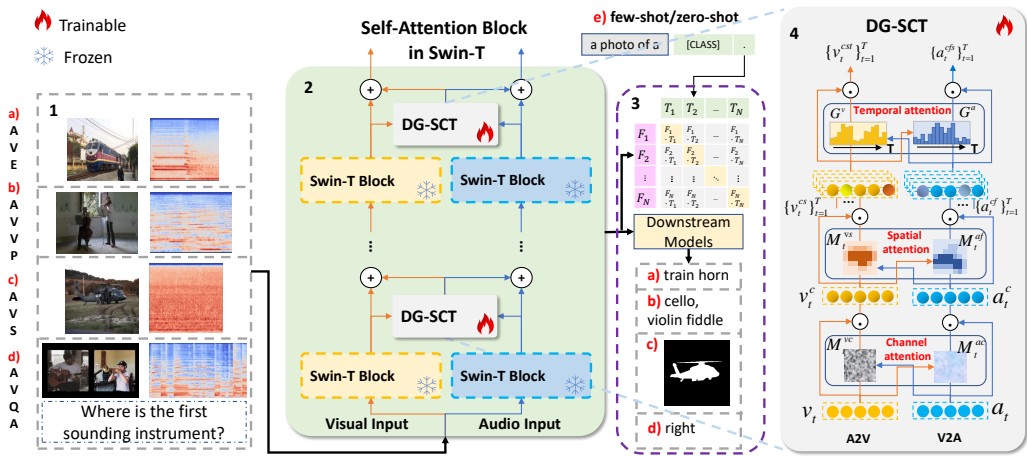

Figure 2: 1) Audio-visual inputs; 2) DG-SCT is injected into every layer of frozen pre-trained audio and visual encoders; 3) After feature extraction, the audio and visual features are sent to various downstream tasks; 4) Details of DG-SCT in spatial-channel-temporal attention levels.

bilities of large-scale models, enabling the effective utilization of task-specific information from different modalities.

- We introduce a novel attention mechanism named Dual-Guided Spatial-Channel-Temporal (DG-SCT), which utilizes audio and visual modalities to guide the feature extraction of their respective counterpart modalities across spatial, channel, and temporal dimensions. Notably, our approach adds only a limited number of parameters for the interaction layer, while keeping the original parameters of the large-scale pre-trained models frozen.

- Extensive experimental results on four audio-visual tasks, namely, AVE, AVVP, AVQA, and AVS, demonstrate the superiority of our model compared to state-of-the-art counterparts across various settings. Furthermore, we evaluate the performance of DG-SCT in few-shot and zero-shot scenarios on the AVE and LLP datasets, demonstrating its superiority over CLIP and several competitive CLIP-Adapters.

## 2 Related work

### 2.1 Audio-visual understanding

Audio-visual understanding tasks involve utilizing both audio and visual modalities to get a better perception of audio-visual scenarios [8, 22, 41, 31]. For instance, **Audio-Visual Event Localization (AVE [33])** requires models to recognize joint audio-visual events. Previous works [33, 16, 35, 37, 36] use late interaction strategies to better leverage the visual and audio features encoded from modality-specific pre-trained models. **Audio-Visual Video Parsing (AVVP [32])** task breaks the restriction that audio and visual signals are definitely aligned. To tackle the weakly-supervised AVVP task, previous work [32] proposes a hybrid attention network and attentive Multimodal Multiple Instance Learning (MMIL) Pooling mechanism to aggregate all features. The task of **Audio-Visual Segmentation (AVS [42])** focuses on whether each pixel corresponds to the given audio so that a mask of the sounding object(s) is generated. Zhou et al. [43] use a temporal pixel-wise audio-visual interaction module to inject audio semantics as guidance for the visual segmentation process. Furthermore, the newly introduced **Audio-Visual Question Answering (AVQA [14])** task requires methods that perceive both audio and visual modalities to answer human-generated questions about the audio-visual content. Li et al. propose a spatiotemporal grounding model [14] to achieve scene understanding and reasoning over audio and visual modalities.

However, most methods designed for these tasks rely on modality-specific audio and visual pre-trained models, which can not utilize multi-modal cues early in the representation stage. In this paper, we propose a novel early-interaction strategy, adaptively extracting key information from the current modality across spatial-channel-temporal dimensions.

## 2.2 Vision-language models and prompt learning

**Vision-language models** have made remarkable progress since the introduction of CLIP [26], with zero-shot and few-shot ideas achieving excellent generalization abilities in many downstream tasks; Meanwhile, **prompt**, a concept in NLP [15, 13, 19], has achieved impressive results in various NLP domains since its introduction, as evidenced by the success of the GPT series [25, 1]. Subsequent works have attempted to combine these two and achieved better results. For example, CoOp [46] improves the CLIP model by optimizing the continuous prompts in the language branch, and CoCoOp [45] further improves the model by incorporating prompts in the video branch. However, these works only utilize prompts to guide individual branches. CLIP-adapter [7] builds on these works by proposing to use embedding of video and language to guide each other at the end of the encoder. MaPLe [12] is the first to use an adaptor to guide each other inside the encoder, integrating visual and text representations with the semantics of each other to enhance the generalization ability of the model. However, none of these works consider utilizing prompts in the audio-visual domain.

In this paper, we introduce a novel bidirectional prompt that employs audio and video cues independent of text to achieve outstanding information extraction abilities for audio-visual tasks.

# 3 Approach

In this section, more details about our proposed DG-SCT are elaborated. An overview of the proposed framework is illustrated in Fig. 2.

## 3.1 Representations for audio-visual modalities

**Visual representation**  Given an input video sequence, we first sample a fixed number of RGB video frames $\{V_t\}_{t=1}^T \in \mathbb{R}^{T \times H \times W \times 3}$, where H, W are height and width. Following the Swin-T [20], we first split each RGB frame $V_t$ into non-overlapping patches by a patch-splitting module with kernel size $(P_v \times P_v)$. Each patch is treated as a "token" and its feature is set as a concatenation of the raw pixel values. A linear embedding layer is then applied to this raw-valued feature and we can get visual features as $v_t \in \mathbb{R}^{\frac{H}{P_v} \times \frac{W}{P_v} \times C_v}$, where $C_v$ is the number of visual channels.

**Audio representation**  Given an input audio track, we first get an audio mel-spectrogram $\{A_t\}_{t=1}^T$, where $A_t \in \mathbb{R}^{L \times F}$ with time $L$ and frequency bins $F$. Following the HTS-AT [3],[3] the audio mel-spectrogram is cut into different patch tokens with a Patch-Embed CNN of kernel size $(P_a \times P_a)$. A linear embedding layer is then applied to this raw-valued feature and we can obtain audio features as $a_t \in \mathbb{R}^{\frac{L}{P_a} \times \frac{F}{P_a} \times C_a}$, where $C_a$ is the number of audio channels.

## 3.2 Adding DG-SCT modules to frozen encoders

Now, we describe how we adjust pre-trained Swin-T and HTS-AT with our proposed DG-SCT. Every layer of the Swin-T and HTS-AT consists of three main operations: 1) multi-head attention (MHA), 2) multi-layer perceptron (MLP), and 3) our DG-SCT modules which use the intermediate layer information of audio and video as prompts to guide each other through spatial-channel-temporal dimensions. We skip the linear normalization layers in both MHA and MLP operations for brevity.

Given audio inputs $a^{(\ell)} \in \mathbb{R}^{T \times (L^{(\ell)} \cdot F^{(\ell)}) \times C_a^{(\ell)}}$ and visual inputs $v^{(\ell)} \in \mathbb{R}^{T \times (H^{(\ell)} \cdot W^{(\ell)}) \times C_v^{(\ell)}}$ from layer $\ell$, we first use a two-dimensional convolution kernel and a linear projection to make the dimensions of the audio and visual prompts consistent of their counterpart modality. Let $v_f^{(\ell)} = \Omega^{\text{a2v}}(a^{(\ell)}, v^{(\ell)})$ and $a_f^{(\ell)} = \Omega^{\text{v2a}}(v^{(\ell)}, a^{(\ell)})$ denote the operation that implements DG-SCT module, which we will describe in the next subsection. Then, the operations in each layer can be written as:

---

[3]`https://github.com/RetroCirce/HTS-Audio-Transformer`, an audio encoder based on Swin-Transformer.

$$v_y^{(\ell)} = v^{(\ell)} + \text{MHA}(v^{(\ell)}) + \Omega^{\text{a2v}}(a^{(\ell)}, v^{(\ell)}), \quad a_y^{(\ell)} = a^{(\ell)} + \text{MHA}(a^{(\ell)}) + \Omega^{\text{v2a}}(v^{(\ell)}, a^{(\ell)}), \quad (1)$$

$$v^{(\ell+1)} = v_y^{(\ell)} + \text{MLP}(v_y^{(\ell)}) + \Omega^{\text{a2v}}(a_y^{(\ell)}, v_y^{(\ell)}), \quad a^{(\ell+1)} = a_y^{(\ell)} + \text{MLP}(a_y^{(\ell)}) + \Omega^{\text{v2a}}(v_y^{(\ell)}, a_y^{(\ell)}).$$
$$(2)$$

## 3.3 Dual-guided spatial-channel-temporal attention

In this subsection, we will describe how DG-SCT works in more detail. Given visual and audio features, the encoder such as Swin-Transformer pre-trained on large-scale single-modal data, will uniformly extract features from audio-visual inputs (See Fig. 1 (a)). However, in practical multi-modal scenarios, not all of this information carries equal importance. For example, as we see in Fig. 1, the region where the car appears in the visual field is evidently more crucial than the background trees. Additionally, the moment when the engine sound emerges in the audio should also be given more attention. Hence, we take advantage of the fact that audio-visual pairs can provide mutual guidance for each other, and utilize different modalities as prompts to help pre-trained models focus more on specific aspects of opposite modal inputs across spatial, channel, and temporal dimensions. Different from previous works [37, 36] which only leverage audio as guidance to extract visual features, our proposed DG-SCT module can achieve triple levels of information highlighting in two directions. We illustrate these cross-modal attention mechanisms in the following parts:

**Channel-wise attention:** Different channels represent different aspects of features. The introduction of channel attention can facilitate the model to ignore irrelevant features and improve the quality of representations [11]. We let the audio and video serve as mutual guidance signals and explicitly model the dependencies between channels on each other's modality. Concretely, We use $\psi_a$ and $\psi_v$ to denote the combinations of convolutional and linear projection in section 3.2, to encode audio and visual inputs as prompts: $a_t' = \psi_a(a_t) \in \mathbb{R}^{C_v \times (H \cdot W)}$ and $v_t' = \psi_v(v_t) \in \mathbb{R}^{C_a \times (L \cdot F)}$, respectively. For audio-to-visual (A2V), we use the spatial average pooling to process $a_t'$ and get $a_t'' \in \mathbb{R}^{C_v \times 1}$, then fuse it with vision via element-wise multiplication after feeding them to projection layers $\Theta_a^c, \Theta_v^c \in \mathbb{R}^{C_v \times C_v}$ respectively, generating audio channel-guidance maps $a_t^{cm} = (\Theta_a^c(a_t'') \odot \Theta_v^c(v_t)) \in \mathbb{R}^{C_v \times (H \cdot W)}$. After that, we spatially squeeze the fused feature by global average pooling, denoted as $\delta_a$, Finally, a bottleneck layer $\Phi_a$ follows with a sigmoid function $\sigma$ is used, yielding channel attention maps $M_t^{vc}$; Similarly, we generate V2A channel attentive maps $M_t^{ac}$:

$$M_t^{vc} = \sigma(\Phi_a(\delta_a(a_t^{cm}))) \in \mathbb{R}^{C_v \times 1}, \quad M_t^{ac} = \sigma(\Phi_v(\delta_v(v_t^{cm}))) \in \mathbb{R}^{C_a \times 1}, \quad (3)$$

where $v_t^{cm} \in \mathbb{R}^{C_a \times (L \cdot F)}$ is the visual channel-guidance maps, $\delta_v$ is spatial-wise global average pooling, $\Phi_v$ indicates a bottleneck layer and $\sigma$ denotes the sigmoid function.

**Spatial-wise attention:** Audio can improve visual feature extraction by contributing to visual attention in the spatial dimension [33]. Inspired by this, we leverage the guidance capabilities of audio and visual prompts to guide visual spatial attention and audio frequency attention, respectively. Similar to the aforementioned channel-wise attention, For A2V, we first get channel-attentive visual features $v_t^c = (M_t^{vc} + 1) \odot v_t$, then we element-wise multiply audio prompt and $v_t^c$ after the projection of $\Theta_a^s$ and $\Theta_v^s$ to hidden dimension $d = 256$, generating audio spatial-guidance maps $a_t^{sm} = (\Theta_a^s(a_t') \odot \Theta_v^s(v_t^c)) \in \mathbb{R}^{d \times (H \cdot W)}$. Then we use a projection layer $\Theta^s \in \mathbb{R}^{1 \times d}$ with a sigmoid function $\sigma$ to obtain spatial attention maps $M_t^{vs}$; Similarly, we generate V2A frequency attentive maps $M_t^{af}$:

$$M_t^{vs} = \sigma(\Theta^s(a_t^{sm})) \in \mathbb{R}^{1 \times (H \cdot W)}, \quad M_t^{af} = \sigma(\Theta^f(v_t^{fm})) \in \mathbb{R}^{1 \times (L \cdot F)}, \quad (4)$$

where $v_t^{fm}$ denotes visual frequency-guidance maps, $\Theta^f \in \mathbb{R}^{1 \times d}$ is a projection layer.

**Temporal-gated attention:** Given an audio, significant time segments (e.g., "engine sound") should be emphasized, while background information (e.g., "silence") should be attenuated. The same holds for the visual information as well [36]. Inspired by this, we add temporal-gated attention in the final layer. For A2V, we first feed the frequency-channel attentive audio features $\{a_t^{cf}\}_{t=1}^{T}$ through an RNN to capture temporal information better and then pass it through a projection layer with

sigmoid function to obtain the temporal attention gates $G^v \in \mathbb{R}^{T \times 1}$; Similarly, for V2A, we feed the spatial-channel attentive visual features to generate $G^a \in \mathbb{R}^{T \times 1}$:

$$G^v = \sigma(\Theta_a^t(\text{RNN}(\{a_t^{cf}\}_{t=1}^T))), \quad G^a = \sigma(\Theta_v^t(\text{RNN}(\{v_t^{cs}\}_{t=1}^T))). \tag{5}$$

**Summary:** We combine the abovementioned three attention mechanisms together to form our DG-SCT module. Given audio features $a_t \in \mathbb{R}^{C_a \times (L \cdot F)}$ and visual features $v_t \in \mathbb{R}^{C_v \times (H \cdot W)}$, DG-SCT first generates channel-wise attention maps $M_t^{vc}$ and $M_t^{ac}$ to let audio and video adaptively emphasize informative features of the corresponding modality. It then lets audio pay attention to the important sounding regions to produce spatial-wise attention maps $M_t^{vs}$ and lets video pay attention to the important frequency regions to generate frequency-wise attention maps $M_t^{af}$, thus the yielding of the spatial-channel attentive visual features $v_t^{cs}$ and the frequency-channel attentive audio features $a_t^{cf}$ can be summarized as:

$$v_t^{cs} = (\alpha \cdot M_t^{vc} + \beta \cdot M_t^{vs} + 1) \odot v_t, \quad a_t^{cf} = (\alpha \cdot M_t^{ac} + \beta \cdot M_t^{af} + 1) \odot a_t, \tag{6}$$

Then we generate two temporal attention gates $G^v \in \mathbb{R}^{T \times 1}$ and $G^a \in \mathbb{R}^{T \times 1}$ for $\{v_t^{cs}\}_{t=1}^T$ and $\{a_t^{cf}\}_{t=1}^T$, respectively, thus the final outputs of $\Omega^{\text{a2v}}$ and $\Omega^{\text{v2a}}$ mentioned in section 3.2 are:

$$\{v_t^{cst}\}_{t=1}^T = (\gamma \cdot G^v + 1) \odot \{v_t^{cs}\}_{t=1}^T, \quad \{a_t^{cft}\}_{t=1}^T = (\gamma \cdot G^a + 1) \odot \{a_t^{cf}\}_{t=1}^T, \tag{7}$$

Where $\alpha$, $\beta$, and $\gamma$ are hyperparameters. Consequently, this approach yields high-quality, fine-grained audio-visual representations, significantly improving the performance of subsequent tasks in the audio-visual domain.

## 4 Experiments

### 4.1 Tasks and datasets

**Audio-visual event localization (AVE)** focuses on recognizing an audio-visual event that is both visible and audible throughout multiple time segments in a video. We evaluate the AVE [33] dataset; **Audio-visual video parsing (AVVP)** aims to parse a video into temporal event segments and label them as either audible, visible, or both. We evaluate our method for weakly-supervised AVVP task on the LLP dataset [32]; The goal of **Audio-visual segmentation (AVS)** is to output a pixel-level map of the object(s) that produce sound at the image frame. We evaluate on AVSBench [43]; **Audio-visual question answering (AVQA)** aims to answer questions based on the associations between objects and sounds. We conduct our experiments on the MUSIC-AVQA dataset [14].

Meanwhile, We propose **Audio-visual few-shot/zero-shot** tasks on AVE [33] and LLP [32] datasets. We evaluate AVE and classification tasks on AVE dataset and classification task on LLP dataset. More details about tasks and datasets will be illustrated in Appendix.

### 4.2 Implementation details

**Audio-visual downstream tasks**: To adapt our approach to the four audio-visual downstream tasks, we replace the pre-trained audio and visual encoders with a frozen HTS-AT [3] and a frozen Swin-Transformer [20], respectively. The trainable DG-SCT modules in Fig. 2 (4) are injected into the frozen layers to let audio and visual modalities guide each other. We then use this as our audio-visual feature extractor. For **AVE** task, our feature extractor is combined with CMBS [36]. The event category label of each video segments is required to be predicted in supervised manner. We adopt [33, 38, 37, 36] and exploit the overall accuracy of the predicted event category as the evaluation metrics; Combined with MGN [24], DG-SCT is able to tackle the weakly-supervised **AVVP** task. Following [32], we evaluate the parsing performance of all events (audio, visual, and audio-visual events) under segment-level and event-level metrics; For **AVS** task, We combine our audio-visual feature extractor with the original AVS model [43]. We use the Jaccard index $\mathcal{J}$ [6] and F-score $\mathcal{F}$

Table 1: **Audio-visual event localization.** Comparisons on the test set of AVE in supervised manner. The result of our re-implementation of LAViSH [17] is 79.7%.

| Method | Visual Encoder | Audio Encoder | Acc |
|---|---|---|---|
| AVEL(Audio-Visual) [33] | VGG-19 | VGG-like | 71.4 |
| AVEL(Audio-Visual+Att) [33] | VGG-19 | VGG-like | 72.7 |
| AVSDN [16] | VGG-19 | VGG-like | 72.6 |
| CMAN [38] | VGG-19 | VGG-like | 73.3 |
| DAM [35] | VGG-19 | VGG-like | 74.5 |
| CMRAN [37] | VGG-19 | VGG-like | 77.4 |
| PSP [44] | VGG-19 | VGG-like | 77.8 |
| CMBS [36] | VGG-19 | VGG-like | 79.3 |
| LAViSH [17] | Swin-V2-L (shared) | | 81.1 |
| LAViSH [17] | Swin-V2-L (shared) | | 79.7 |
| LAViSH*[4] | Swin-V2-L | HTS-AT | 78.6 |
| **Ours** | Swin-V2-L | HTS-AT | **82.2** |

Table 2: **Audio-visual video parsing.** Comparisons on the test set of LLP.

| Methods | Segment-level | | | | | Event-level | | | | |
|---|---|---|---|---|---|---|---|---|---|---|
| | A | V | AV | Type | Event | A | V | AV | Type | Event |
| AVE [33] | 49.9 | 37.3 | 37.0 | 41.4 | 43.6 | 43.6 | 32.4 | 32.6 | 36.2 | 37.4 |
| AVSDN [16] | 47.8 | 52.0 | 37.1 | 45.7 | 50.8 | 34.1 | 46.3 | 26.5 | 35.6 | 37.7 |
| HAN [32] | 60.1 | 52.9 | 48.9 | 54.0 | 55.4 | **51.3** | 48.9 | 43.0 | 47.7 | 48.0 |
| MGN [24] | **60.7** | 55.5 | 50.6 | 55.6 | **57.2** | 51.0 | 52.4 | 44.4 | 49.3 | **49.2** |
| **Ours** | 59.0 | **59.4** | **52.8** | **57.1** | 57.0 | 49.2 | **56.1** | **46.1** | **50.5** | 49.1 |

as the evaluation metrics; For **AVQA** task, our audio-visual feature extractor is used in the original ST-AVQA [14]. Similarly, we use the answer prediction accuracy as the evaluation metrics.

**Audio-visual few-shot/zero-shot tasks**: We incorporate DG-SCT modules as adapters between the frozen CLIP image encoder ViT [26] and frozen CLAP audio encoder HTS-AT [5], generating audio and visual features for text-audio and text-image contrastive learning, as shown in Fig. 2 (e). For the zero-shot setting, the model is pre-trained on the VGG-Sound(40K) [2] dataset. More details will be discussed in Appendix.

### 4.3 Compared with state-of-the-arts on audio-visual downstream tasks

First, we challenge our method against current state-of-the-art methods on the four audio-visual tasks. As demonstrated in Table 1, our model outperforms CMBS [36] and LAViSH [17] by a significant margin (**2.9%** and **2.5%**); In Table 2, our model attains either a competitive or even better performance. For instance, DG-SCT surpasses MGN by **3.9%** points in the segment-level visual event parsing metric, demonstrating our method can utilize large-scale models to extract more useful features and further promote the fusion of these features in downstream tasks. We also achieve state-of-the-art results on S4 setting of AVS task (Table 3). Lastly, our model performs exceptionally well on AVQA task and outperforms previous leading methods on AQ, VQ, and AVQ question types, respectively. The experimental results reveal that our model has the capability of utilizing pre-trained audio and visual models to extract more comprehensive, higher-quality features tailored for downstream tasks than the cross-attention mechanism of LAViSH. Moreover, our model exhibits excellent generalization abilities, achieving impressive results across various audio-visual tasks.

### 4.4 Compared with state-of-the-arts on audio-visual few-shot/zero-shot tasks

Furthermore, as presented in Fig. 4, our DG-SCT model surpasses top-performing methods like CoCoOp, CLIP-Adapter, and MaPLe by a non-negligible margin on our newly proposed audio-visual few-shot/zero-shot scenarios. Our few-shot (shot=16) learning for the AVE task attains **72.4%**,

---

[4]Our modified implementation of LAViSH for fair comparisons.

Table 3: **Audio-visual segmentation.** Comparisons on the S4 and MS3 settings of AVSBench. $\mathcal{M}_\mathcal{J}$ and $\mathcal{M}_\mathcal{F}$ denote the mean $\mathcal{J}$ and $\mathcal{F}$ metric values (section 4.2) over the whole dataset.

| Method | Visual Encoder | Audio Encoder | Setting | | | |
|---|---|---|---|---|---|---|
| | | | S4 | | MS3 | |
| | | | $\mathcal{M}_\mathcal{J}$ | $\mathcal{M}_\mathcal{F}$ | $\mathcal{M}_\mathcal{J}$ | $\mathcal{M}_\mathcal{F}$ |
| AVS [43] | PVT-v2 | VGG-like | 78.7 | 87.9 | **54.0** | **64.5** |
| LAVisH [17] | Swin-V2-L (shared) | | 80.1 | 88.0 | 49.8 | 60.3 |
| LAVisH* | Swin-V2-L | HTS-AT | 78.0 | 87.0 | 49.1 | 59.9 |
| **Ours** | Swin-V2-L | HTS-AT | **80.9** | **89.2** | 53.5 | 64.2 |

Table 4: **Audio-visual question answering.** Comparisons on the test set of MUSIC-AVQA. We report accuracy on three types of questions, e.g., audio (AQ), visual (VQ), and audio-visual (AVQ).

| Method | Visual Encoder | Audio Encoder | AQ | VQ | AVQ | Avg |
|---|---|---|---|---|---|---|
| AVSD [28] | VGG-19 | VGG-like | 68.5 | 70.8 | 65.5 | 67.4 |
| Pano-AVQA [40] | Faster RCNN | VGG-like | 70.7 | 72.6 | 66.6 | 68.9 |
| ST-AVQA [14] | ResNet-18 | VGG-like | 74.1 | 74.0 | 69.5 | 71.5 |
| LAVisH [17] | Swin-V2-L(shared) | | 75.7 | 80.4 | 70.4 | 74.0 |
| LAVisH* | Swin-V2-L | HTS-AT | 75.4 | 79.6 | 70.1 | 73.6 |
| **Ours** | Swin-V2-L | HTS-AT | **77.4** | **81.9** | **70.7** | **74.8** |

outperforming MaPLe for **5.3%** points, and is even comparable to previous full-training baselines (Table 1). Note that experimental results for few-shot on the LLP classification task are worse than zero-shot scenario. We argue that this is due to a significant gap between downstream and pre-training data, which may have disrupted the original parameters of the CLIP model. However, we still outperform MaPLe by **2.2%** and **0.7%** points, for zero-shot and few-shot (shot=16) settings, respectively. The outcomes demonstrate that our dual-guided audio-visual prompts extract task-specific information better than text-image [26, 46, 45, 7, 12] and text-audio [5] pre-trained models and exhibit stronger adaptability on audio-visual tasks.

## 4.5 Ablation analysis

We verify the effectiveness of the three modules, namely, **S** (spatial), **C** (channel), and **T** (temporal), in **DG-SCT**. As shown in Table 5, compared to the baseline model, the individual incorporation of these attention modules leads to substantial performance enhancements of our model across a range of downstream tasks, demonstrating the effectiveness of these modules. Although cross-modal interactions in any dimension can enhance feature extraction, we argue that distinct modules can also mutually guide and enhance each other's performance. For instance, removing the **T** module individually does not result in a significant decrease in the performance of the model, which indicates that the combination of **S** and **C** can slightly substitute the function of **T**. The reason might be that if the features of a certain segment are more prominent in both spatial and channel dimensions, it is more likely to be important in the temporal dimension as well.

Table 5: Ablation study of the spatial-channel-temporal attention module. "S", "C", and "T" denote spatial, channel, and temporal attention, respectively.

| Module | | | AVE | AVVP | | AVS | | | | AVQA | | | |
|---|---|---|---|---|---|---|---|---|---|---|---|---|---|
| S | C | T | Acc | Segment-level AV | Event-level AV | S4 | | MS3 | | AQ | VA | AVQ | Avg |
| | | | | | | $\mathcal{M}_\mathcal{J}$ | $\mathcal{M}_\mathcal{F}$ | $\mathcal{M}_\mathcal{J}$ | $\mathcal{M}_\mathcal{F}$ | | | | |
| - | - | - | 78.6 | 49.8 | 43.9 | 78.0 | 87.0 | 49.1 | 59.9 | 75.4 | 79.6 | 70.1 | 73.6 |
| - | ✓ | - | 81.8 | 51.3 | 45.6 | 79.9 | 88.4 | 51.0 | 62.1 | 76.0 | 80.9 | 70.8 | 74.4 |
| ✓ | - | - | 80.9 | 51.7 | 44.7 | 78.6 | 87.6 | 50.9 | 61.6 | 74.9 | 80.3 | 69.5 | 73.3 |
| ✓ | ✓ | - | 82.0 | 52.3 | 45.9 | 80.0 | 88.6 | 51.8 | 62.3 | 77.0 | 81.9 | 70.3 | 74.6 |
| - | - | ✓ | 81.5 | 50.9 | 45.7 | 79.9 | 88.9 | 52.2 | 61.5 | 76.0 | 81.3 | 70.2 | 74.1 |
| ✓ | ✓ | ✓ | **82.2** | **52.8** | **46.1** | **80.9** | **89.2** | **53.5** | **64.2** | **77.4** | **81.9** | **70.7** | **74.8** |

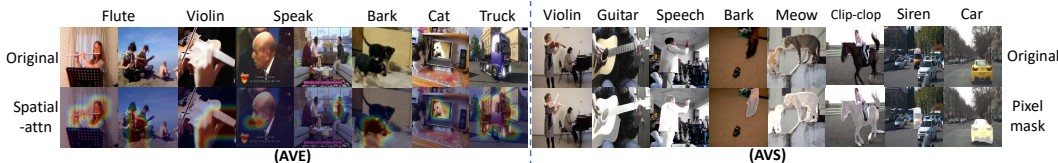

Figure 3: The qualitative results of DG-SCT on AVE and AVS tasks.

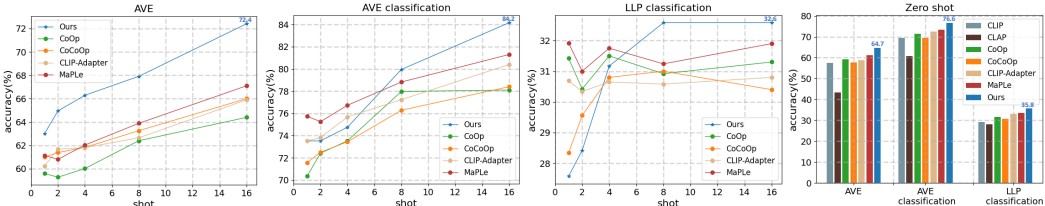

Figure 4: Results of our model and previous methods on **few-shot/zero-shot tasks.**

## 4.6 Qualitative analysis

Fig. 3 represents examples of the effectiveness of our DG-SCT. On the left, we observe that with the guidance of the audio prompts, the module can accurately focus on the critical visual regions for different AVE events. For instance, when multiple objects are present in video frames, our model is capable of accurately localizing the sounding objects (e.g., bark, cat, the second example of flute, the fifth example of speak). Moreover, our model can precisely pinpoint the sound source location at a fine-grained level, such as the strings of a violin and the hands of the performer, and the speaker's mouth. In addition, DG-SCT achieves excellent results on AVS task. The right part of Fig. 3 indicates that our model can accurately segment the pixels of sounding objects and outline their shapes perfectly. The excellent qualitative results of our model on various downstream tasks illustrate its strong potential for generalization.

As depicted in Fig. 5, we employ t-SNE [34] to visualize the learned audio and visual features and compare features generated by our model with features generated by baseline without DG-SCT, on various tasks. Each spot denotes the feature of one audio or visual event, and each color corresponds to a particular category, such as "cat" in orange as shown in Fig. 5 (AVE). As we can see, features extracted by the proposed DG-SCT are more intra-class compact and more inter-class separable. These meaningful visualizations further demonstrate that the DG-SCT model successfully learns compact and discriminative features for each modality across diverse downstream tasks.

## 4.7 Efficiency analysis

Our efficiency analysis on AVE task is presented in Table 6. Our approach utilizes more trainable parameters. However, our proposed DG-SCT attention mechanism requires a comparable number of parameters to the latent tokens utilized in LAVisH [17]. The increase trainable parameters primarily arises from the inclusion of a two-dimensional convolution kernel and a linear projection (section 3.2). These additions ensure consistency of dimensions between the audio and visual prompts. In other words, the increase in parameter count from our approach mainly results from addressing the inconsistency in dimensions between the audio and visual encoders.

We also conducted a comparison of computational cost [29] on AVE task. Our approach involves fine-tuning the pre-trained model, which inevitably leads to a reduction in speed compared to previous late-interaction baselines (CMBS [36]). However, we have achieved outstanding results, and our approach is applicable to multiple audio-visual tasks. Overall, the benefits are substantial.

## 5 Conclusion and discussion

This paper introduces DG-SCT, a method that leverages audio and visual modalities as prompts in the early layers of frozen pre-trained encoders. By doing so, our model can extract higher-quality

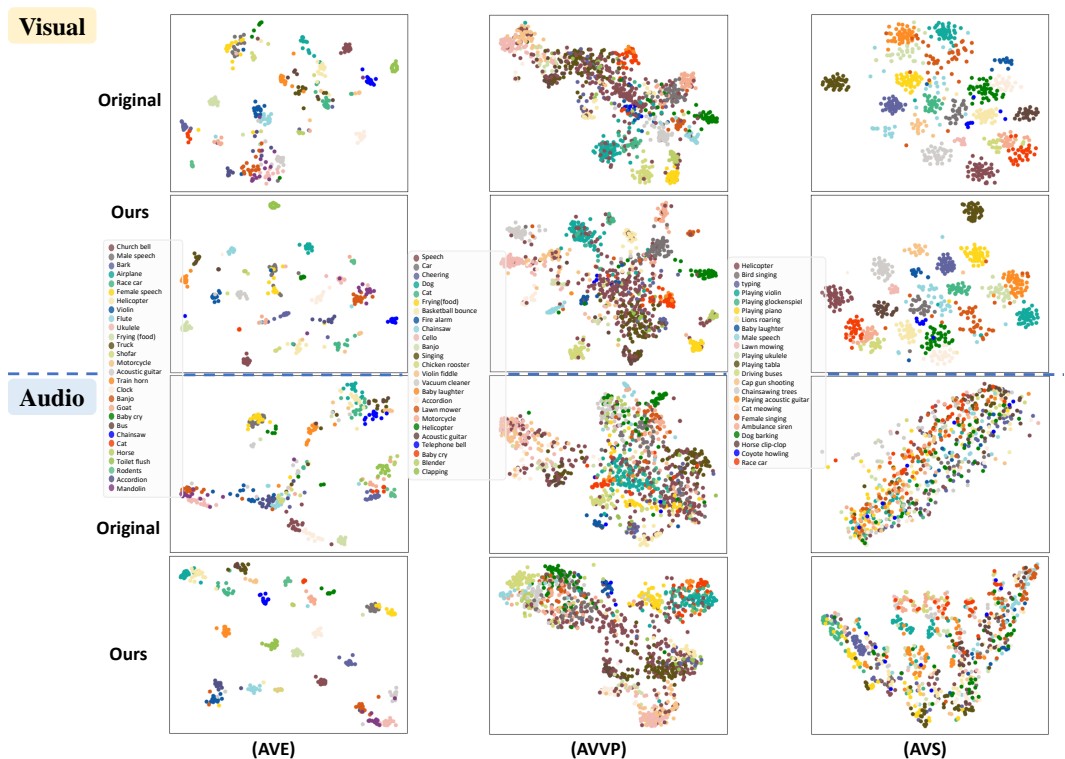

Figure 5: Qualitative visualizations of original visual (top row), our visual (second row), original audio (third row), and our audio (bottom row) features on AVE, AVVP, and AVS tasks.

Table 6: **Efficiency analysis on AVE task.** For trainable parameters of LAVisH* and our model, the first number (17.3) represents the trainable parameters of a two-dimensional convolution kernel and a linear projection.

| Method | Trainable Params (%) | Total Params (M) | GFLOPs | Acc |
|---|---|---|---|---|
| CMBS [36] | 14.4 | 216.7 | 40.9 | 79.3 |
| LAVisH [17] | 10.1 | 238.8 | 406.7 | 79.7 |
| LAVisH* | 17.3+**13.3**=30.6 | 374.9 | 416.1 | 78.6 |
| **Ours** | 17.3+**26.3**=43.6 | 461.3 | 460.8 | **82.2** |

and finer-grained audio-visual features, enhancing performance in subsequent tasks. We conduct comprehensive experiments on four datasets, including AVE, AVVP, AVS, and AVQA, as well as our newly proposed few-shot/zero-shot audio-visual tasks. Across **25** experimental settings, our approach achieves state-of-the-art results on **19** of them. Additionally, ablation studies conducted on these datasets validate the effectiveness of our proposed spatial, channel, and temporal attention modules. Furthermore, our approach demonstrates robust generalizability and holds potential for application in more audio-visual scenarios in the future.

# 6 Acknowledgements

This work is supported by National Key R&D Program of China under Grant No.2022ZD0162000, National Natural Science Foundation of China under Grant No. 62222211 and No.61836002. We also gratefully acknowledge the support of MindSpore (https://www.mindspore.cn), which is a new deep learning computing framework.

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

# A Implementation details

For the **AVE** and the **AVQA** tasks, we set $\alpha = 0.3$, $\beta = 0.05$, and $\gamma = 0.1$. We train the model with a batch size of 8 and a learning rate of $5 \times 10^{-4}$ and $1 \times 10^{-4}$, respectively; For the **AVVP** task, we set $\alpha = 0.3$, $\beta = 0.05$, and $\gamma = 0.05$. We train the model with a batch size of 8 and a learning rate of $3 \times 10^{-4}$; For the **S4** setting of the **AVS** task, we set $\alpha = 0.3$, $\beta = 0.05$, and $\gamma = 0.05$. We train the model with a batch size of 8 and a learning rate of $3 \times 10^{-4}$; For the **MS3** setting of the **AVS** task, we set $\alpha = 0.2$, $\beta = 0.1$, and $\gamma = 0.1$. We train the model with a batch size of 2 and a learning rate of $1.5 \times 10^{-4}$.

For **few-shot/zero-shot tasks**, we set the learning rate to $3 \times 10^{-4}$ with a batch size of 2. For the **AVE** task, we set $\alpha = 0.2$, $\beta = 0.05$, and $\gamma = 0.01$; For the **AVE classification** and the **LLP classification** tasks, we set $\alpha = 0.2$, $\beta = 0.05$, and $\gamma = 0.05$.

All of our experiments are trained on one NVIDIA A100 GPU.

# B More details of datasets

Table 7: **Dataset statistics.** Each dataset is shown with the number of videos and the *annotated* frames. The "annotations" column indicates whether the frames are labeled by category, pixel-level masks, or answer.

| Datasets | Videos | Frames | Classes/ Answers | Types | Annotations |
|---|---|---|---|---|---|
| AVE [33] | 4,143 | 41,430 | 28 | video | category |
| LLP [32] | 11,849 | 11,849 | 25 | video | category |
| AVSBench [43] | 5,356 | 12,972 | 23 | video | pixel |
| VGG-Sound(40K) [2] | 40,801 | 408,010 | 141 | video | category |
| MUSIC-AVQA [14] | 9,288 | 45,867 | 42 | video | answer |

**AVE dataset** [5] We evaluate the AVE task on the AVE dataset [33] originating from the AudioSet [9]. The AVE dataset contains $4,143$ videos covering 28 categories. Each video lasts for 10 seconds and contains an event category labeled for each video on a segment level.

**LLP dataset** [6] We evaluate the AVVP task on the LLP dataset [32], which has $11,849$ video-level event annotations on the presence or absence of different video events and each video is $10s$ long and has at least $1s$ audio or visual events. There are $7,202$ videos that contain events from more than one event category and per video has averaged $1.64$ different event categories. For evaluation, there are $4,131$ audio events, $2,495$ visual events, and $2,488$ audio-visual events for the $1,849$ videos.

**AVSBench dataset** [7] We evaluate the AVS task on the AVSBench dataset [43], which contains two settings: 1) Single Sound Source Segmentation (S4), which contains $4,932$ videos over 23 categories; 2) Multiple Sound Source Segmentation (MS3), which contains 424 videos over 23 categories. Each video is 5 seconds long.

**MUSIC-AVQA dataset** [8] We conduct our experiments of the AVQA task on the MUSIC-AVQA dataset [14], which contains $9,288$ videos, $45,867$ question-answer pairs, 33 question templates, and 42 answers.

**VGG-Sound(40K)** [9] The pre-training data of our zero-shot task is VGG-Sound(40K), which is split from the VGG-Sound dataset [2]. VGG-Sound contains over 200k clips for 300 different sound classes, while our VGG-Sound(40K) has $40,801$ clips for 141 sound classes.

---

[5] https://github.com/YapengTian/AVE-ECCV18
[6] https://github.com/YapengTian/AVVP-ECCV20
[7] http://www.avlbench.opennlplab.cn/download.
[8] https://gewu-lab.github.io/MUSIC-AVQA/.
[9] https://www.robots.ox.ac.uk/~vgg/data/vggsound/.

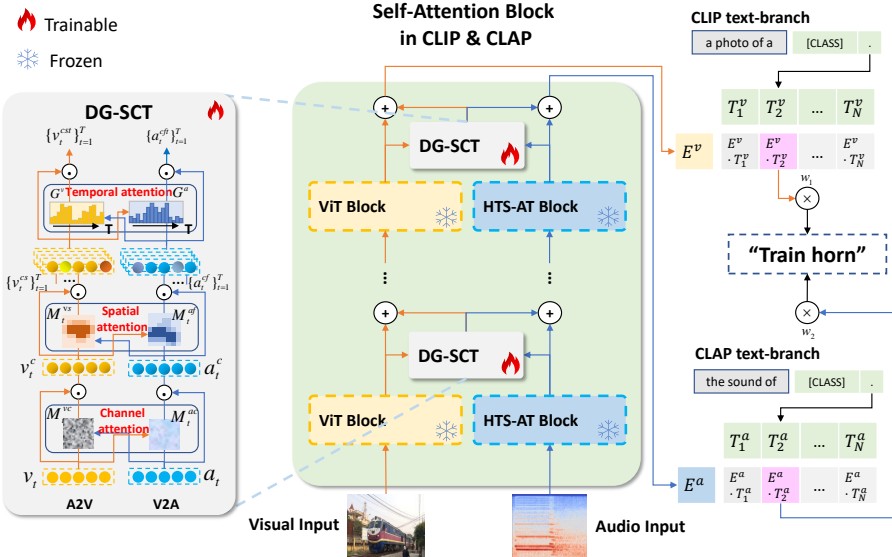

Figure 6: Details of our framework in few-shot/zero-shot scenarios. **Right:** two text branches are used to operate image-text and audio-text matching, respectively. The final result is obtained by adding the weighted sum of the two outputs from image-text matching and audio-text matching.

## C  More details of few-shot/zero-shot tasks

### C.1  Framework and method

As shown in Fig. 6, our DG-SCT module is injected into every layer of the frozen CLIP [26] image encoder and the frozen CLAP [5] audio encoder. After that, the image encoder and the audio encoder are denoted as $f_v(\cdot)$ and $f_a(\cdot)$, respectively. Two text branches $f_t^v(\cdot)$, $f_t^a(\cdot)$ are employed to perform image-text and audio-text matching, respectively, as the text encoders from different pre-trained models exhibit a substantial gap, using only one of them will result in a significant decrease in accuracy.

Let $(v_i, a_i, t_i)$ represents a piece of visual-audio-text pair indexed by $i$. The visual embedding $E_i^v$, the audio embedding $E_i^a$, and the corresponding text embeddings $T_i^v$ and $T_i^a$, are obtained by encoders with projection layers, respectively:

$$E_i^v = \mathrm{MLP}_v(f_v(v_i)), \quad E_i^a = \mathrm{MLP}_a(f_a(a_i)), \quad T_i^v = \mathrm{MLP}_t^v(f_t^v(t_i)), \quad T_i^a = \mathrm{MLP}_t^a(f_t^a(t_i)), \quad (8)$$

where the visual/audio/text projection layer is a 2-layer multilayer perception (MLP) with ReLU as the activation function to map the encoder outputs into the same dimension $D$ (i.e., $E_i^v, E_i^a, T_i^v, T_i^a \in \mathbb{R}^D$).

The model is trained with the contrastive learning paradigm between the paired visual and text embeddings, as well as the paired audio and text embeddings, following the same loss function in [26]:

$$\mathcal{L}_v = \frac{1}{2N} \sum_{i=1}^{N} (\log \frac{\exp(E_i^v \cdot T_i^v / \tau_v)}{\sum_{j=1}^{N} \exp(E_i^v \cdot T_j^v / \tau_v)} + \log \frac{\exp(T_i^v \cdot E_i^v / \tau_v)}{\sum_{j=1}^{N} \exp(T_i^v \cdot E_j^v / \tau_v)}), \quad (9)$$

$$\mathcal{L}_a = \frac{1}{2N} \sum_{i=1}^{N} (\log \frac{\exp(E_i^a \cdot T_i^a / \tau_a)}{\sum_{j=1}^{N} \exp(E_i^a \cdot T_j^a / \tau_a)} + \log \frac{\exp(T_i^a \cdot E_i^a / \tau_a)}{\sum_{j=1}^{N} \exp(T_i^a \cdot E_j^a / \tau_a)}), \quad (10)$$

Where $\tau_v$ and $\tau_a$ are learnable temperature parameters for scaling the loss. $N$ is usually the number of data, but during the training phase, $N$ is used as the batch size. Let $y_v$ denotes the image-text

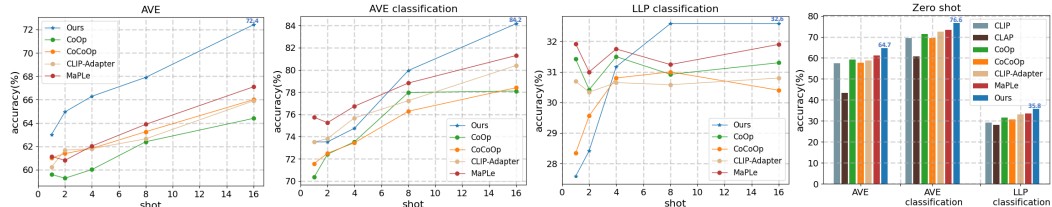

Figure 7: Results of our model and previous methods on **few-shot/zero-shot tasks.**

matching score, and $y_a$ denotes the audio-text matching score, we set hyperparameters $w_1 = \frac{y_v}{y_v + y_a}$ and $w_2 = 1 - w_1$, thus the overall objective function is:

$$\mathcal{L} = w_1 \cdot \mathcal{L}_v + w_2 \cdot \mathcal{L}_a, \tag{11}$$

When the image-text matching score is higher, it indicates that the visual modality is more certain in determining the category of the sample, therefore the model needs to increase the weight of visual modality in the loss function, and vice versa.

The same weight settings for $w_1$ and $w_2$ in Fig. 6 are also significant during the inference process, indicating that some samples require more guidance from the audio modality while others might require more guidance from the visual modality.

For the few-shot setting, we train the model by selecting $shot$ samples of each category from the training dataset of the downstream tasks. For the zero-shot setting, we pre-train the model using the VGG-Sound(40K) dataset.

### C.2   Evaluation metrics

For the **AVE** task, the category label of each video segment is required to be predicted in supervised manner. We adopt [33, 37, 36], and exploit the overall accuracy of the predicted event category as the evaluation metrics; For the **AVE classification** task, the category label of each video is required to be predicted. We leverage the overall accuracy of the video category as the evaluation metrics; For the **LLP classification** task, the category label of each video is required to be predicted. Note that some videos might have multiple categories, we randomly select one as ground truth for brevity. We leverage the overall accuracy of the video category as the evaluation metrics.

Table 8: Ablation study of A2V and V2A modules on AVE, AVS, and AVQA tasks.

| Module | | AVE | AVS | | | | AVQA | | | |
| A2V | V2A | Acc | S4 | | MS3 | | AQ | VQ | AVQ | Avg |
| | | | $\mathcal{M}_{\mathcal{J}}$ | $\mathcal{M}_{\mathcal{F}}$ | $\mathcal{M}_{\mathcal{J}}$ | $\mathcal{M}_{\mathcal{F}}$ | | | | |
|---|---|---|---|---|---|---|---|---|---|---|
| - | - | 78.6 | 78.0 | 87.0 | 49.1 | 59.9 | 75.4 | 79.6 | 70.1 | 73.6 |
| ✓ | - | 79.2 | 80.7 | 89.0 | 51.4 | 61.6 | 76.1 | **82.0** | 70.8 | 74.7 |
| - | ✓ | 81.3 | 79.1 | 88.0 | 50.7 | 62.4 | 75.9 | 80.8 | **70.8** | 74.3 |
| ✓ | ✓ | **82.2** | **80.9** | **89.2** | **53.5** | **64.2** | **77.4** | 81.9 | 70.7 | **74.8** |

### C.3   Results

As shown in Fig. 7 (a), (b) and (c), for **few-shot** ($shot = 16$) setting, our method outperforms MaPLe by a significant margin on AVE, AVE classification, and LLP classification tasks, achieving **72.4%**, **84.2%**, and **32.6%**, respectively. However, the performance of our model on both the AVE and LLP classification tasks may not be as effective as some previous methods when $shot < 8$. The classification tasks are relatively simple and are already closely related to the image-text matching task in the upstream pre-training of the CLIP model. Therefore, the CLIP model's pre-trained parameters can achieve good results. However, when $shot > 8$, our model gains significant advantages. We also observe that the accuracy curves of other few-shot learning methods are relatively flat, particularly for

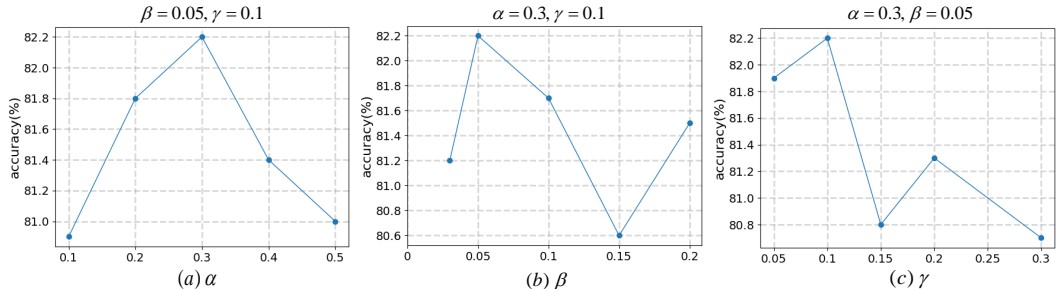

Figure 8: **Hyperparameters.** We explore the impact of the hyperparameters $\alpha$, $\beta$, and $\gamma$ on the experimental results of the AVE task. a) Fix $\beta = 0.05$ and $\gamma = 0.1$, observe the impact of $\alpha$; b) Fix $\alpha = 0.3$ and $\gamma = 0.1$, observe the impact of $\beta$; c) Fix $\alpha = 0.3$ and $\beta = 0.05$, observe the impact of $\gamma$.

the LLP classification task, where the accuracy sometimes decreases when $shot$ increases. This issue might be the limited ability of the image-text models to extract rich information from the training data of audio-visual tasks. Thus, the potential for improving accuracy is limited. For **zero-shot** setting, our model also achieves state-of-the-art on all three tasks. The results demonstrate that our model can extract audio-visual task information more effectively and achieve better results.

Table 9: Ablation study of A2V and V2A modules on AVVP task.

| Module | | Segment-level | | | | | Event-level | | | | |
|---|---|---|---|---|---|---|---|---|---|---|---|
| A2V | V2A | A | V | AV | Type | Event | A | V | AV | Type | Event |
| - | - | 57.8 | 56.3 | 49.8 | 55.2 | 54.9 | 48.2 | 51.7 | 43.9 | 48.8 | 47.6 |
| ✓ | - | 56.4 | **59.5** | **53.3** | 56.4 | 55.0 | 47.4 | 55.9 | **46.3** | 49.9 | 47.8 |
| - | ✓ | **59.4** | 57.3 | 50.8 | 55.8 | 56.8 | 49.2 | 54.1 | 44.4 | 49.2 | 48.6 |
| ✓ | ✓ | 59.0 | 59.4 | 52.8 | **57.1** | **57.0** | **49.2** | **56.1** | 46.1 | **50.5** | **49.1** |

# D    More ablation studies

Next, we conduct more ablation studies to investigate how different components of our model affect the performance on the downstream tasks.

**A2V and V2A modules.** We first investigate the effectiveness of the bidirectional attention mechanism. We compare our final bidirectional approach with the unidirectional variants that only use either audio-to-visual (A2V) or visual-to-audio (V2A) spatial-channel-temporal attention mechanism, and also a baseline that does not use any cross-modal connections.

As indicated in Table 8, integrating A2V or V2A individually leads to substantial performance enhancements across AVE, AVS, and AVQA tasks compared to the baseline model. Furthermore, the bidirectional DG-SCT outperforms the unidirectional A2V and V2A baselines (e.g., $3.0\%$ and $0.9\%$ on the AVE task). In Table 9, we observe that using the A2V module alone does not significantly decrease the accuracy for visual and audio-visual events. However, without visual guidance (V2A), the performance of audio events suffers a considerable decline; Likewise, the performance of visual events drops without audio guidance (using the V2A module alone). These experimental findings demonstrate the necessity of visual guidance for audio events and the need for audio guidance for visual events. Our proposed DG-SCT model can bidirectionally guide the representation of each modality, thus enhancing the accuracy of downstream audio-visual tasks.

**Hyperparameters.** Now, we explore the impact of the hyperparameters $\alpha$, $\beta$, and $\gamma$ on the experimental results of the AVE task. As we can see in Fig. 8, our model achieves the best accuracy (**82.2%**) when $\alpha = 0.3$, $\beta = 0.05$, and $\gamma = 0.1$.

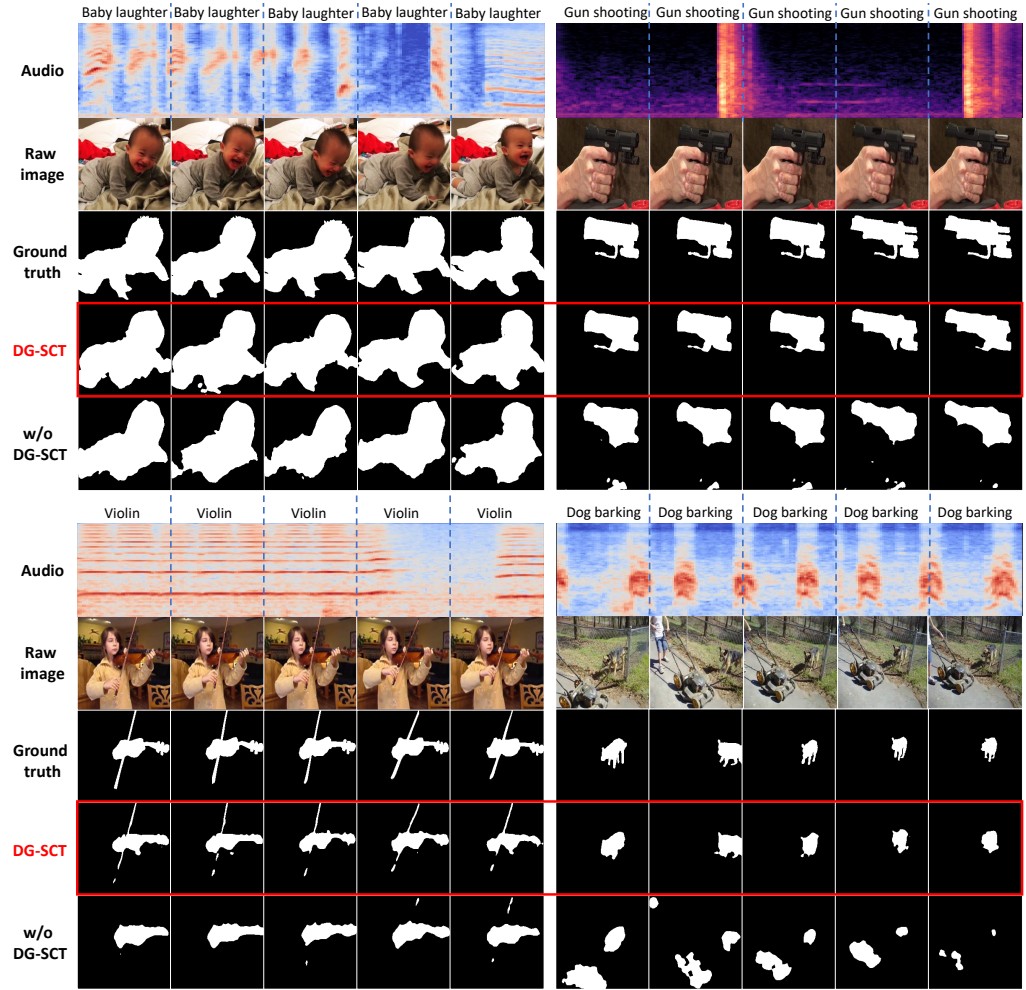

Figure 9: **Qualitative examples of the baseline method (w/o DG-SCT) and our DG-SCT framework,** under the S4 setting of the AVS task.

## E  Additional qualitative analyses

**Qualitative examples of the AVS task.** We provide some qualitative examples of the AVS task to test the effectiveness of the DG-SCT model. As shown in Fig 9, under the S4 setting, the DG-SCT model can not only outline the object shapes more perfectly (shown in the upper left, upper right, and bottom left figures) but also locate the sounding objects more accurately than the baseline model without the DG-SCT module. As we can see in the bottom right figure of Fig. 9, DG-SCT can locate the dog that is barking, simultaneously excluding the mower that hasn't produced sound. The baseline model, however, mistakenly locates the dog and the mower at the same time. In Fig 10, under the MS3 setting, the DG-SCT model can locate multiple sound sources and outline their shape nicely. In some difficult cases (the bottom row in Fig. 10), DG-SCT can still pinpoint the correct sounding source even if the sound has stopped or the scene has changed dramatically.

**Qualitative examples of the AVVP task.** In Fig. 11, we visualize the video parsing results of DG-SCT and baseline (w/o DG-SCT) on different examples. "GT" denotes the ground truth annotations. The results show that adding the DG-SCT module achieves more accurate parsing performance by acquiring mutual guidance from audio and visual modalities during the representation. For example, in Fig. 11 (b), our model can recognize the mixture of "banjo" and "singing" while the baseline model (w/o DG-SCT) falsely predicts the sound as "violin fiddle".

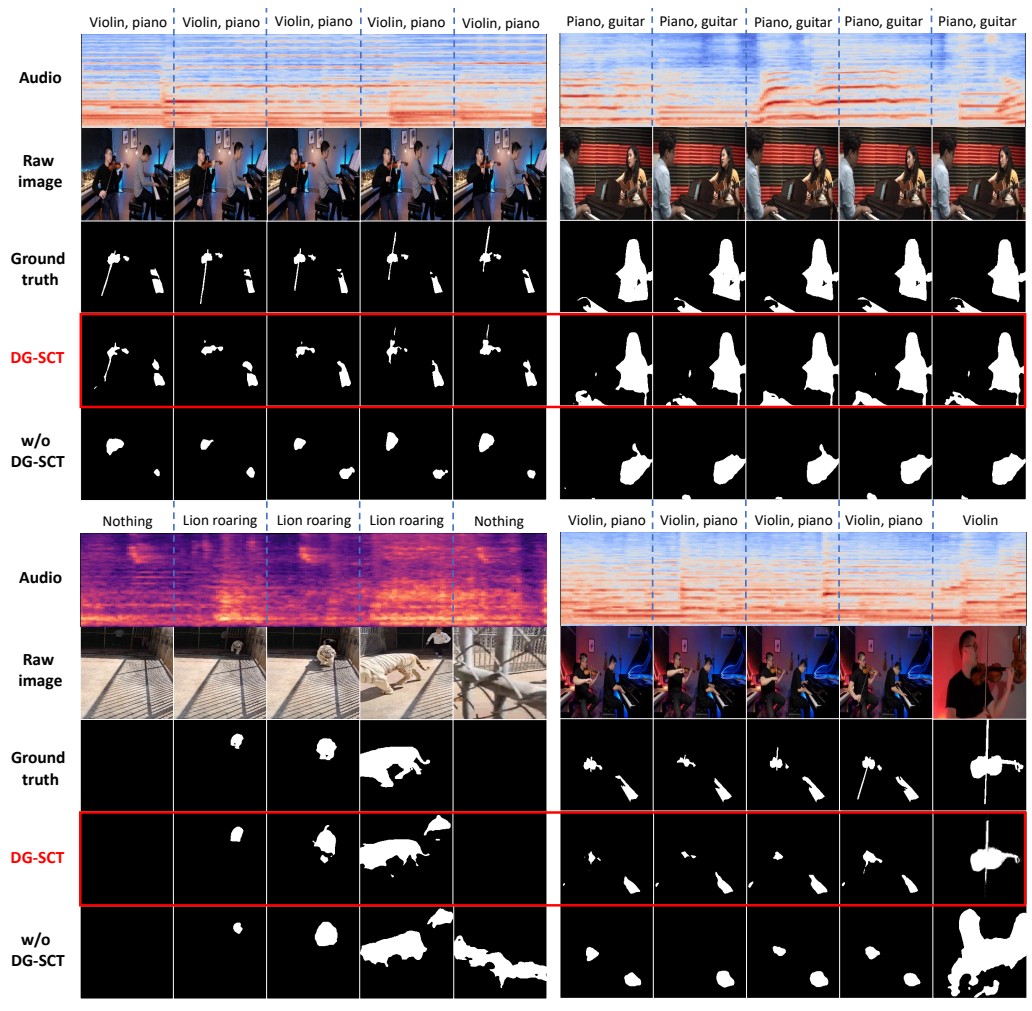

Figure 10: **Qualitative examples of the baseline method (w/o DG-SCT) and our DG-SCT framework,** under the MS3 setting of the AVS task.

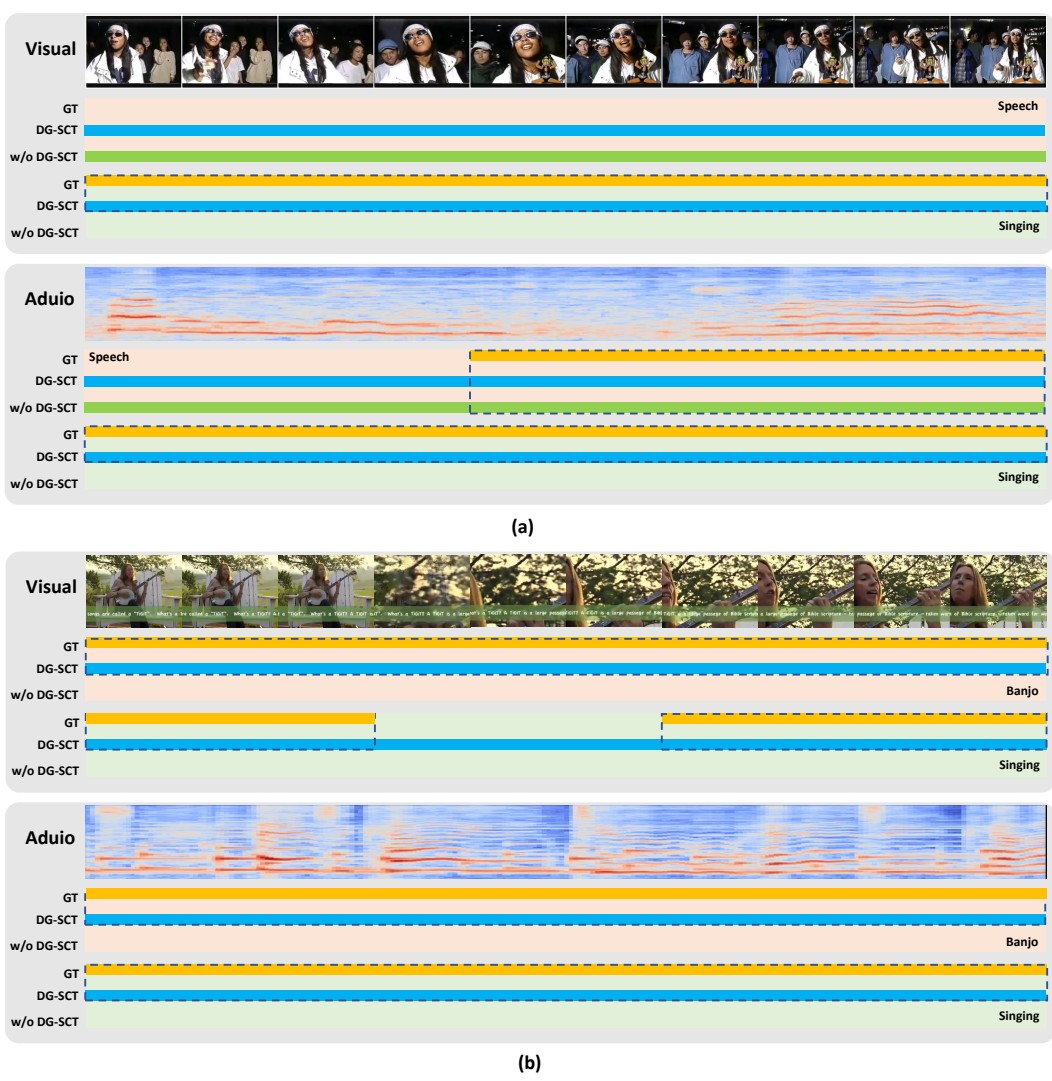

Figure 11: **Qualitative examples of the baseline method (w/o DG-SCT) and our DG-SCT framework,** under the AVVP task.

