# OpenReview forum: "Cross-modal Prompts: Adapting Large Pre-trained Models for Audio-Visual Downstream Tasks"
_NeurIPS.cc/2023/Conference — NeurIPS 2023 poster_

### Official Review · Reviewer_TYiB · 2023-07-03

**Soundness:** 3 good
**Presentation:** 2 fair
**Contribution:** 3 good
**Rating:** 6
**Confidence:** 4

**Summary:**

The authors propose a novel dual-guided spatial-channel-temporal attention mechanism to audio-visual problems, which leverages pre-trained audio and visual encoders. And they show the improvement in various audio-visual tasks such as event localization, parsing, segmentation, and question answering.

**Strengths:**

- This work applied proposed methods and evaluated with diverse audio-visual downstream tasks, including localization, parsing, segmentation and question answering. This helps to showcase the generalization of proposed approach.

**Weaknesses:**

- The use of notations in Section 3.3 is very complicated and difficult to follow. Consider utilize Figure 2 (4) for illustration and walk through each step along with the modules in the figure to make it easier for the readers to follow.
- For the results presented in Table 2 and 3, and discussed in Section 4.3, only the better performance is highlighted without providing potential explanation and/or hypothesis why proposed system performed worse in some scenarios. For example, the performance on event-level as in Table 2, and the performance with MS3 of segmentation in Table 3. Consider adding some discussion for these cases.
- Some abbreviations are referred to without any information provided, such as LLP in line 210, and CMBS in line 218. Consider adding one-liner explanation to help the reader.
- Minor comments: Section 3.2 in equation (1), the first superscript "l" does not follow the same style as others.

**Questions:**

- Do you need to use the same number of layers in the Swin-T and HTS-AT? Since they share the same notation in all the equations for both audio and visual paths.
- In Section 3.1, line 127, where does T come from? Could you provide more detail on the choices of mel-spectrogram?

**Limitations:**

- No potential social or ethical implications.

---

> ### Author Rebuttal · Authors · 2023-08-09
>
> Dear Reviewer TYiB,
>
> Thank you so much for giving us positive feedback and the very insightful questions. Please see our point by point responses below.
>
> ------
>
> **Weakness 1 - The use of notations in Section 3.3 is very complicated and difficult to follow.**
>
> Thank you for your suggestion. We will revise it. After revising it like this, we will consider adding this passage to the new paper:
>
> 3.3 ...., our proposed DG-SCT module can achieve triple levels of information highlighting in two directions **(See Fig. 2 (4))**. ....
> **Channel-wise attention:** .... Similarly, we generate V2A channel attentive maps $M_t^{ac}$
>
>  **(See Fig. 2 (4))**. ....
>
> **Spatial-wise attention:** .... Similarly, we generate V2A frequency attentive maps $M_t^{af}$
>
> **(See Fig. 2 (4))**. ....
>
> **Temporal-gated attention:** .... Similarly, for V2A, we feed the spatial-channel attentive visual features to generate $G^a$
>
> **(See Fig. 2 (4))**. ....
>
> **Summary:** .... We combine the abovementioned three attention mechanisms together to form our DG-SCT module. **Walking through each step along with Fig. 2 (4).**.....
>
> ------
>
> **Weakness 2 - Only the better performance is highlighted without providing potential explanation and/or hypothesis why proposed system performed worse in some scenarios.**
>
> Thank you for your concern. We will add more explanation for the results presented in Table 2 and 3, and discussed in Section 4.3 in the future:
>
> In the experimental results presented in Table 2 and Table 3, we observed that for the scenes where our performance was not higher, it is comparable to the state-of-the-art results reported in previous studies. However, a notable decline is observed in the audio scene depicted in **Table 2**. Based on our analysis, it is plausible that the lower robustness of HTS-AT, compared to previous audio encoders such as VGG-like, may be a contributing factor to this decline.
>
> ------
>
> **Weakness 3 - Some abbreviations are referred to without any information provided.**
>
> Thank you for your suggestion. LLP refers to Look, Listen, and Parse dataset for audio-visual video scene parsing. CMBS refers to Cross Modal Background Suppression for Audio-Visual Event Localization. We have already provided some details about the datasets and tasks in **Appendix B**, specifically on **line 15**. However, we acknowledge that we overlooked the explanation of abbreviations in our previous discussion. Additionally, we will provide a concise introduction to the abbreviations in our Appendices.
>
> ------
>
> **Weakness 4 - Section 3.2 in equation (1), the first superscript "l" does not follow the same style as others.**
>
> We would like to extend our sincere appreciation for your dedicated effort in carefully reviewing our work! This issue has been addressed.
>
> ------
>
> **Question 1 - Do you need to use the same number of layers in the Swin-T and HTS-AT?**
>
> Thank you for your insightful question, we are so glad that you asked that. No, we don't need to use the same number of layers, or the same size of dimensions. To address the dimension asynchrony problem, **"we first use a two-dimensional convolution kernel and a linear projection to make the dimensions of the audio and visual prompts consistent of their counterpart modality", line 139**. For full-training settings for AVE/AVS/AVVP/AVQA tasks, although both Swin-V2-L and HTS-AT has **4** layers, the numbers of the blocks of each layer are not the same, with **2, 2, 18, 2** and **2, 2, 6, 2**, respectively. We designed an approach that the layers with inconsistent block counts **(e.g., the third layer, 18 and 6, respectively)** engage as many rounds of cross-modal interactions as possible **(e.g., 6)**, uniformly.
>
> So you see, the number of layers or the size of dimensions does not impose any limitations. Our approach is applicable to any transformer architecture.
>
> ------
>
> **Question 2 - In Section 3.1, line 127, where does T come from? Could you provide more detail on the choices of mel-spectrogram?**
>
> Thank you for asking. We split the mel-spectrogram into **T** non-overlapping segments, each segment is **1** second long. Spectrograms, in simple terms, involve dividing an audio waveform into small windows. Each window undergoes a short-time discrete Fourier transform to obtain frequency and amplitude information; As for Mel Scale, mathematically speaking, is the result of some non-linear transformation of the frequency scale. This Mel Scale is constructed such that sounds of equal distance from each other on the Mel Scale, also “sound” to humans as they are equal in distance from one another. The mel-scale is a logarithmic scale that captures the perceptual sensitivity of human hearing to different frequencies. Mel-spectrogram, which leverages the mel-scale for effective characterization of audio signals, is commonly used in audio representation.
>
> Thank you for your feedback. We hope the aforementioned clarifies any doubts or inquiries you may have had.

---

> > ### Comment · Reviewer_TYiB · 2023-08-20
> >
> > Thank for the authors for your responses and clarifications. I would suggest integrating these clarification into the final paper if possible. I am maintaining my rating.

---

> > > ### Author Response · Authors · 2023-08-20
> > >
> > > Dear Reviewer TYiB,
> > >
> > > Thank you so much for checking our response. We are glad that you keep the original positive rating.

---

### Official Review · Reviewer_myp2 · 2023-07-04

**Soundness:** 2 fair
**Presentation:** 2 fair
**Contribution:** 2 fair
**Rating:** 4
**Confidence:** 5

**Summary:**

This paper proposes a parameter-efficient approach, DG-SCT. DG-SCT can adapt pre-trained audio and visual models on downstream audio-visual tasks without updating pre-trained encoders (i.e., keep pre-trained encoders frozen.)

**Strengths:**

$+$ DG-SCT can achieve state-of-the-art results on several downstream audio-visual tasks.
$+$ DG-SCT is able to learn channel-wise, spatial, and temporal information to incorporate with pre-trained audio and visual encoders.

**Weaknesses:**

$-$ Although DG-SCT leverages different types of cross-modal attention (i.e., channel-wise, spatial, and temporal), the design is not clear.
For example, DG-SCT leverages RNN for modeling cross-modal temporal information and learnable weights for cross-modal spatial information. Such a technique can be implemented with divided (space-time) attention after channel-wise attention.

$-$ The effectiveness of the proposed temporal modeling is not clear. The baselines for the implementation of DG-SCT in AVE/AVVP/AVQA have already cross-modal temporal modeling modules.

$-$ The number of trainable parameters is not reported. For example, the baselines in AVVP and AVE usually use pre-extracted audio and visual features, which contribute to a lower number of trainable parameters.

$-$ Lack of efficiency comparison (e.g., FLOPs). Combining several attention mechanisms will lead to huge computational costs. It would be great to include these metrics.


$-$ some experimental settings and results are not clear (See. Questions)

**Questions:**

* In AVE task, CMBS uses much weaker audio and visual encoders. How does CMBS with Swin-V2-L and HTS-AT work? Why does CMBS drop from 79.3 to 78.6 in Table 5?

* In AVVP task, the results of MGN in Table 2 are lower than the numbers in the paper. Does DG-SCT use [A], which is also used in MGN?

* Similar to the question in AVE task,  how does AVS with Swin-V2-L and HTS-AT work in AVS task? (the performance is also dropped in Table 5)

* Are the baselines in zero/few-shot tasks also pre-trained in VGGSound (40K)? If yes, that would be fair to compare with DG-SCT.

[A] Wu, Yu, and Yi Yang. "Exploring heterogeneous clues for weakly-supervised audio-visual video parsing." CVPR 2021

**Limitations:**

Yes, DG-SCT uses more number of trainable parameters

---

> ### Author Rebuttal · Authors · 2023-08-09
>
> Dear Reviewer myp2,
>
> Thank you so much for taking the time to read our paper and providing very insightful and constructive comments and questions, please see the following for our point-by-point reply. We also conducted quite many new experiments to address your concerns.
>
> ------
>
> **Weaknesses 1 - Such a technique can be implemented with divided (space-time) attention after channel-wise attention.**
>
> We understand the reviewer's point. The reason we leverages RNN for modeling cross-modal temporal information is that, in  **line 180**, **Given an audio, significant time segments (e.g., "engine sound") should be emphasized, while background information (e.g., "silence") should be attenuated. The same holds for the visual information as well.**
>
> We made an effort to incorporate divided (space-time) attention following channel-wise attention, but regrettably, the obtained results were not encouraging. This can be attributed to the absence of guidance from the other modality, which plays a pivotal role in audio-visual scenarios.
>
> ------
>
> **Weaknesses 2 - The effectiveness of the proposed temporal modeling is not clear.**
>
> Thank you for asking! We understand you concern. In **4.5 Ablation analysis, line 263,** though we acknowledged that **"removing the 'T' module individually does not result in a significant decrease in the performance of the model"** because **"if features of a certain segment are more prominent in both spatial and channel dimensions, it is more likely to be important in the temporal dimension as well"**, adding the "T" module (temporal modeling) did make a difference. For example, after adding our "T" module, the results raised from **51.8** and **62.3** to **53.5** and **64.2**,  respectively,  in the MS3 setting of AVS task, even though the downstream model already has temporal modeling for late interaction.
>
> ------
>
> **Weaknesses 3 - The  number of trainable parameters is not reported.**
>
> Thank you for your suggestion. We apologize for omitting the information about the number of trainable parameters in our initial submission. We will include it in our Appendix section. Please find the updated information below (AVE task):
>
> | Method  | Trainable Params (M) | Total Params (M) | Acc      |
> | ------- | -------------------- | ---------------- | -------- |
> | CMBS    | 14.4                 | 216.7            | 79.3     |
> | LAVisH  | 10.1                 | 238.8            | 79.7     |
> | LAVisH* | 30.6                 | 374.9            | 78.6     |
> | Ours    | 43.6                 | 461.3            | **82.2** |
>
> In "LAVisH*",  we modified LAVisH to use the same encoders as ours for fair comparison.
>
> In contrast to retraining the complete backbones, we have effectively achieved a substantial reduction in the number of trainable parameters. Furthermore, our approach has demonstrated a remarkable improvement in performance compared to other baselines while increasing acceptable parameter count.
>
> ------
>
> **Weaknesses 4 - Lack of efficiency comparison (e.g., FLOPs).**
>
> We thank the reviewer for pointing this out. We acknowledge that our previous consideration may have been inadequate, since the audio-visual task is not inherently time-sensitive, and our previous baselines did not report FLOPs information. Here, we will add the efficiency comparison on AVE task in Appendices:
>
> | Method  | GFLOPs | Acc      |
> | ------- | ------ | -------- |
> | CMBS    | 40.9   | 79.3     |
> | LAVisH  | 406.7  | 79.7     |
> | LAVisH* | 416.1  | 78.6     |
> | Ours    | 460.8  | **82.2** |
>
> In "LAVisH*",  we modified LAVisH to use the same encoders as ours for fair comparison.
>
> Our approach involves fine-tuning the pre-trained model, which unavoidably results in a decrease in speed compared to the previous late-interaction baselines (CMBS). However, we have achieved excellent results and our approach is applicable to multiple audio-visual tasks. Overall, the benefits are substantial.
>
> ------
>
> **Question 1 and 3 - Settings and results in AVE task and AVS task.**
>
> We thank the reviewer for the valuable input. The original LAVisH baseline uses Swin-L-V2 to encode both audio and visual modalities. For fair comparison, we modified LAVisH with Swin-L-V2 and HTS-AT, same as ours, to encode visual and audio modalities, respectively. The reason why the results dropped to **78.6** may be attributed to the rudimentary approach of LAVisH, which utilizes latent tokens for cross-attention. The coarse extracted cross-modal information may not adequately counteract the negative effects of domain gaps introduced by the use of different encoders.
>
> ------
>
> **Question 2 - Settings and results in AVVP task.**
>
> Thank you for your question. No, we didn't use [A] in MGN because MGN does not open source the code about using [A]. The results correspond to MGN paper without using [A].
>
> ------
>
> **Question 4 - Are the baselines in zero/few-shot tasks also pre-trained on VGGSound (40k)?**
>
> Thank you for asking! This is a very insightful question. Yes, the baselines are also pre-trained on VGGSound (40K).

---

> > ### Comment · Reviewer_myp2 · 2023-08-14
> >
> > Thanks for providing the response. I partially address my concern. However, weaknesses 3 and Question 1 are still remaining.
> >
> > 1. DG-SCT uses more trainable parameters than LAVISH (parameter-efficient method) and CMBS (conventional method).
> > 2. More issues come from the table in weaknesses 3. For example, CMBS uses fewer trainable parameters and weaker pre-trained audio and visual encoders.
> > 3. LAVisH* may not be implemented correctly. If the authors just replace the pre-trained audio and visual encoders for LAVISH, the trainable parameters should be similar to 10.1M.
> >
> > Overall, I think the comparison is completely fair. Thus, I keep my rating the same

---

> > > ### Author Response · Authors · 2023-08-17
> > >
> > > Dear Reviewer myp2,
> > >
> > > Thank you so much for checking our response. Let us respond to your questions point by point.
> > >
> > > **Question 1 and 3 - DG-SCT uses more trainable parameters; LAVisH\* may not be implemented correctly.**
> > >
> > > Thank you for your question. Yes, our approach utilizes more trainable parameters. However, our proposed Dual-guided spatial-channel-temporal attention mechanism requires a comparable number of parameters to the latent tokens utilized in LAVisH. The increase in trainable parameters mainly stems from the inclusion of a two-dimensional convolution kernel and a linear projection. These additions ensure the consistency of dimensions between the audio and visual prompts.
> > >
> > > | Method  | Trainable Params (M) | Total Params (M) | Acc      |
> > > | ------- | -------------------- | ---------------- | -------- |
> > > | CMBS    | 14.4                 | 216.7            | 79.3     |
> > > | LAVisH  | 10.1                 | 238.8            | 79.7     |
> > > | LAVisH* | 17.3+**13.3**=30.6   | 374.9            | 78.6     |
> > > | Ours    | 17.3+**26.3**=43.6   | 461.3            | **82.2** |
> > >
> > > In "LAVisH*",  we modified LAVisH to use the same encoders as ours for fair comparison.
> > >
> > > In Column Trainable Params (M), Row "LAVisH*" and Ours, the first number (17.3) represents the trainable parameters of a two-dimensional convolution kernel and a linear projection.
> > >
> > > In other words, the increase in parameter count from our approach primarily stems from the inconsistency in dimensions between the audio and video encoders. Unlike LAVisH, which utilizes the same encoder for both audio and video, the dimension inconsistency in our method leads to a higher number of trainable parameters. This also addresses question 3 -- Yes, LAVisH* is implemented correctly. Apart from addressing the dimension inconsistency between the audio and visual encoders, the parameters of LAVisH* (13.3M) and LAVisH (10.1M) are comparable.
> > >
> > > Our model demonstrates strong versatility. It performs effectively across multiple tasks and shows significant improvements compared to LAVisH (a parameter-efficient method), which is relatively poorer on the AVVP task and AVS MS3 task, and CMBS (a conventional method), which can only be used in AVE task.
> > >
> > > Here are the results showcasing the AVVP and AVS MS3 tasks:
> > >
> > > | Method | Segment-level |          |          |          |          | Event-level |          |          |          |          |
> > > | :----: | :-----------: | -------- | -------- | -------- | -------- | ----------- | -------- | -------- | -------- | -------- |
> > > |        |       A       | V        | AV       | Type     | Event    | A           | V        | AV       | Type     | Event    |
> > > |  HAN   |     60.1      | 52.9     | 48.9     | 54.0     | 55.4     | **51.3**    | 48.9     | 43.0     | 47.7     | 48.0     |
> > > |  MGN   |   **60.7**    | 55.5     | 50.6     | 55.6     | **57.2** | 51.0        | 52.4     | 44.4     | 49.3     | **49.2** |
> > > | LAVisH |     57.2      | 54.3     | 50.4     | 54.9     | 56.3     | 47.4        | 51.2     | 45.1     | 49.0     | 48.5     |
> > > |  Ours  |     59.0      | **59.4** | **52.8** | **57.1** | 57.0     | 49.2        | **56.1** | **46.1** | **50.5** | 49.1     |
> > >
> > > | Method | Mj       | Mf       |
> > > | ------ | -------- | -------- |
> > > | AVS    | **54.0** | **64.5** |
> > > | LAVisH | 49.8     | 60.3     |
> > > | Ours   | 53.5     | 64.2     |
> > >
> > > It can be observed that LAVisH fails to achieve satisfactory performance in challenging tasks such as AVVP and AVS MS3. In many settings, it even lags behind the previous baseline. This is attributed to the coarse extraction of cross-modal information through the latent tokens. Our method, however, achieves state-of-the-art or comparable results.
> > >
> > > **Question 2 - More issues come from the table in weaknesses 3. For example, CMBS uses fewer trainable parameters and weaker pre-trained audio and visual encoders.**
> > >
> > > We thank the reviewer for pointing this out. As mentioned above, the increase in parameter count from our approach primarily stems from the inconsistency in dimensions between the audio and visual encoders. As for the "weaker pre-trained audio and visual encoders" problem, the LAVisH paper has already replaced the encoders of CMBS for fair comparison. By replacing the visual encoder with Swin-V2-L, CMBS achieves an accuracy of **80.4%** on the AVE task. Therefore, it is justifiable to claim that our method (**82.2%**) represents the state-of-the-art.
> > >
> > > Furthermore, our method pioneers the better application of pre-training models in the multimodal domain.

---

> > > > ### Comment · Reviewer_myp2 · 2023-08-19
> > > >
> > > > Thanks for the responses. It partially addressed my concerns. However, a fair baseline of LAVISH* should be replaced swin-v2 for audio encoder with HTS-AT. The trainable parameters should be similar to LAVISH or even less (because HTS-AT has fewer layers).
> > > >
> > > > The current LAVISH* may include the proposed method and a better pre-trained audio encoder (v.s. no pre-trained audio encoder for LAVISH). It is difficult to evaluate baselines.

---

> > > > > ### Author Response · Authors · 2023-08-19
> > > > >
> > > > > Dear Reviewer myp2,
> > > > >
> > > > > Thank you so much for your reply.
> > > > >
> > > > > ------
> > > > >
> > > > > In light of your statement, "a fair baseline of LAVISH* should be replaced swin-v2 for audio encoder with HTS-AT," we completely agree with this perspective and that is exactly what we have done. We intentionally refrained from including our proposed method (DG-SCT) or a pre-trained audio encoder that surpasses HTS-AT.
> > > > >
> > > > > Regarding your question, we actually addressed it in our previous response. However, we understand that our explanation may not have been sufficiently clear: **Both LAVisH\* and our approach need a two-dimensional convolution kernel and a linear projection to insure the consistency of dimensions between the audio and visual modalities.** The reason is that **both the latent token method in LAVisH and our DG-SCT mechanism require the audio and video to have the same dimensions**.
> > > > >
> > > > > | Method  | Trainable Params (M) | Total Params (M) | Acc      |
> > > > > | ------- | -------------------- | ---------------- | -------- |
> > > > > | CMBS    | 14.4                 | 216.7            | 79.3     |
> > > > > | LAVisH  | 10.1                 | 238.8            | 79.7     |
> > > > > | LAVisH* | 17.3+**13.3**=30.6   | 374.9            | 78.6     |
> > > > > | Ours    | 17.3+**26.3**=43.6   | 461.3            | **82.2** |
> > > > >
> > > > > In "LAVisH*",  we modified LAVisH to use the same encoders as ours (Swin-V2-L for video, HTS-AT for audio) for fair comparison.
> > > > >
> > > > > In Column Trainable Params (M), Row "LAVisH\*" and Ours, **the first number (17.3) represents the trainable parameters of a two-dimensional convolution kernel and a linear projection.**
> > > > >
> > > > > ------
> > > > >
> > > > > In other words, the increase in parameter count from LAVisH\* primarily stems from the inconsistency in dimensions between the audio and video encoders. Unlike LAVisH, which utilizes the same encoder for both audio and video, **the dimension inconsistency in LAVisH\* leads to a higher number of trainable parameters**. Excluding the parameters related to dimension consistency handling, the trainable parameters of **LAVisH\* (13.3M)** are comparable to those of **LAVisH (10.1M)**.
> > > > >
> > > > > ------
> > > > >
> > > > > We hope that the aforementioned response adequately addresses your question. We would be greatly appreciative if you could reassess our work and consider adjusting the score.

---

### Official Review · Reviewer_VeZU · 2023-07-07

**Soundness:** 3 good
**Presentation:** 3 good
**Contribution:** 3 good
**Rating:** 6
**Confidence:** 3

**Summary:**

This work proposes a new mechanism to utilize audio-visual features as novel prompts to extract task-specific features from large-scale models. This work introduces an attention mechanism named Dual-Guided Spatial-Channel-Temporal (DG-SCT), which utilizes audio and visual modalities to guide the feature extraction of their respective counterpart modalities across spatial, channel, and temporal dimensions. The proposed method is evaluated on a series of tasks including Audio-visual event localization, Audio-visual video parsing, Audio-visual segmentation, and Audio-visual question answering. Moreover, it proposes a new benchmark to perform Audio-visual few-shot/zero-shot tasks on AVE and LLP datasets.


**Strengths:**

- This is a very interesting work. The use of prompting is primarily focused on language and later vision, notably this work performs audio-visual prompting, which seems quite innovative.
-  Moreover, unlike previous works that offer unidirectional prompts, the proposed approach introduces bidirectional prompts, where both visual and audio modalities can mutually guide each other in the feature extraction process.
- The proposed approach is evaluated on several benchmarks and compared fairly with prior works showing the effectiveness of the proposed method.
- It's a nicely written paper and easy to follow.

**Weaknesses:**

Please see #Questions for more open-ended discussions.



**Questions:**

1. The proposed mechanism more resembles with Adapters than Prompts. Prompts usually work in the input space, while the proposed DG-SCT mechanism is also added in the intermediate layers. Moreover, prompting generally assumes no access to the models' internal architecture, but the proposed mechanism needs the models' access to modify/adjust some of the intermediate layers to connect the DG-SCT module.

2. Often audio and video representations suffer from substantial domain gaps due to their modality-specific nature, did you encounter such scenarios?

3. The proposed method is based on the Swin transformer, could you please elaborate if your method can be adapted to other Transformer architectures (e.g., other ViT variants)? If so, what necessary changes need to be done or is it a Swin-specific solution?

4. In this study, the pretrained encoders are trained in a uni-modal setup, do you think the proposed solution is equally effective if multimodal pretrained networks are used? Could you please run an experiment on such a setup using any state-of-the-art AV Transformer model?

5. Did you try validating your method on modality-agnostic backbones or can you show results on such setup? modality-agnostic backbones refer to when the same network is trained with both modalities, e.g., VATT (https://arxiv.org/abs/2104.11178), XKD (https://arxiv.org/pdf/2211.13929.pdf).

**Limitations:**

Please see #Questions.
My questions are mostly open-ended. I will look forward to the discussion with the authors and the arguments of the other reviewers.

---

> ### Author Rebuttal · Authors · 2023-08-09
>
> Dear Reviewer VeZU,
>
> Thank you for the positive feedback and all these very valuable and constructive questions/suggestions! Let us respond to your questions point by point.
>
> ------
>
> **Question 1 - The proposed mechanism more resembles with Adapters than Prompts.**
>
> Thank you very much for you kind suggestion! We initially referred to our proposed mechanism as ‘prompts’ to represent the guidance provided to the learnable weights in preceding layers. It emphasizes using audio and video as prompts to guide the representation of the counterpart modalities. However, we acknowledge that this expression may have been imprecise. Upon considering your valuable suggestion, we will name it as ‘adapters’ instead. We sincerely appreciate your valuable input and assure you that we will carefully consider incorporating this change.
>
> ------
>
> **Question 2 - Did you encounters domain gaps?**
>
> We thank the reviewer for pointing this out. The reviewer is very correct that domain gaps exist in the field of multi-modal tasks due to the unique characteristics of each modality. **For full-training settings** like AVE/AVS/AVVP/AVQA tasks, during the model designing process, we carefully considered employing methods such as contrastive learning to reduce the gaps between modalities. However, we observed limited effectiveness in addressing this issue. The reason is that our proposed method focuses on fine-tuning a minimal number of parameters in the encoder of the current modality using prompt information derived from the counterpart modality. The majority of parameters remain frozen, thereby effectively mitigating the domain gap problem while enriching the modality information. After late interaction and fusion in the downstream models, our method is able to achieve promising results. **For zero-shot/few-shot scenarios**, as mentioned in **line 228, our method "generates audio and visual features for text-audio and text-image contrastive learning"**. Here we use two text branches instead of one **(See Appedices Figure 1 "Right")** to address the domain gaps of visual and audio modalities.
>
> ------
>
> **Question 3 - Could you please elaborate if your method can be adapted to other Transformer architectures?**
>
> Thank you for asking this meaningful question! **Our method is not a Swin-specific solution.** As mentioned in **4.2 Implementation details, line 227**, in our few-shot/zero-shot scenarios, **"We incorporate DG-SCT modules as adapters between the frozen CLIP image encoder ViT and frozen CLAP audio encoder HTS-AT"**. Our method can be adapted to other transformer variants as well. Since different transformer architectures may have different dimensions and layers. First, as mentioned in **line 139**, **"we first use a two-dimensional convolution kernel and a linear projection to make the dimensions of the audio and visual prompts consistent of their counterpart modality"**. For full-training settings like AVE/AVS/AVVP/AVQA tasks, although both Swin-V2-L and HTS-AT has **4** layers, the numbers of the blocks of each layer are not the same, with **2, 2, 18, 2** and **2, 2, 6, 2**, respectively. We designed an approach that the layers with inconsistent block counts **(e.g., the third layer, 18 and 6, respectively)** engage as many rounds of cross-modal interactions as possible **(e.g., 6)**, uniformly.
>
> So you see, the number of layers or the size of dimensions does not impose any limitations. Our approach is applicable to any transformer architecture.
>
> ------
>
> **Question 4 and 5 - Do you think the proposed solution is equally effective if multimodal pretrained networks are used?**
>
> Regarding the incompatibility issue between the official VATT code, which is implemented in TensorFlow, and our PyTorch version, as well as the unavailability of open-source code for XKD, we have indeed encountered some challenges. However, we have identified a recent study on modality-agnostic backbones called meta-transformer. Please see the following table for the results on AVE task:
>
> | Method                       | Acc      |
> | ---------------------------- | -------- |
> | LAVisH                       | 79.7     |
> | LAVisH*                      | 78.6     |
> | Ours                         | **82.2** |
> | LAVisH with meta-transformer | 74.8     |
> | Ours with meta-transformer   | 77.5     |
>
> In "LAVisH*",  we modified LAVisH to use the same encoders as ours for fair comparison.
>
> However, the results were not ideal. Merely replacing the baseline encoders with the modality-agnostic backbone of the meta-transformer resulted in significant performance loss. This indicates that current modality-agnostic backbones may not perform well in the domain of audio-visual downstream tasks. Nevertheless, incorporating our method into the modality-agnostic backbone still yields performance improvements, highlighting the robustness of our approach.
>
> We express our gratitude to the reviewer for highlighting this aspect. Indeed, the utilization of a modality-agnostic backbone appears to be a highly promising trend. We will continue to stay updated with relevant research endeavors and anticipate its eventual suitability for audio-visual downstream tasks in the future.

---

### Official Review · Reviewer_BzrV · 2023-07-26

**Soundness:** 3 good
**Presentation:** 3 good
**Contribution:** 3 good
**Rating:** 6
**Confidence:** 4

**Summary:**

This paper mainly proposes a new attention mechanism named Dual-Guided Spatial-Channel-Temporal (DG-SCT), which utilizes audio and visual modalities to guide the feature extraction of their respective counterpart modalities across spatial, channel, and temporal dimensions. Experiments on 4 tasks shows the advantage of the proposed method.

**Strengths:**

Overall it is a nice paper.

- The presentation of this paper is very clear.

- Experiments are extensive and appear solid.

- From a high level, I think a better audio-visual attention mechanism of this type would benefit a series of downstream tasks. Prompting is a trend to make audio-visual systems smarter.



**Weaknesses:**

With the above said, I am not sure if the claim on Page 1 lines 25-27 is valid for all audio-visual tasks. "However, when perceiving the roaring sound of an engine, the visual region depicting a "car" should receive more attention than the region of "trees". Simultaneously, when observing the car, it is crucial to concentrate on the audio segments of the engine sound." it is true for tasks about audio-visual correspondence like retrieval/localization/segmentation, etc. But in other tasks like audio-visual joint classification, we do want to leverage the information that uniquely appears in a single modality to make predictions. This is because if we only use mutual information, then a single modality is enough, what we are looking for is information not appear in one modality but can be found in the other one. The proposed method seems to attend to the mutual information. I am wondering if it would negatively impact the performance of joint classification tasks.

- minor: Page 2, line 54, it should be HTS-AT, not HT-SAT.

**Questions:**

For the evaluation tasks, it would be nice to have a joint classification task such as AudioSet classification.

**Limitations:**

There is one sentence limitation "We consume a few more parameters than LAVisH." I don't think this weakens the proposed method.

---

> ### Author Rebuttal · Authors · 2023-08-09
>
> Dear Reviewer BzrV,
>
> Thank you for taking the time to consider our paper and giving us positive feedback!
>
> ------
>
> Regarding your question, **"I am wondering if it would negatively impact the performance of joint classification tasks."**
>
> It is a very good question. In joint classification task, mutual information is critical as well. For example, when considering only the visual modality, a radio appears motionless. However, by taking into account the audio, one can determine whether the radio is producing sound. Similarly, when only listening to the audio, the sound of turning on and off a light may sound similar. It is only by combining the visual information that we can determine whether the light is on or off. By leveraging the information from both modalities, a more comprehensive and complete understanding can be achieved.
>
> It seems you are referring to the consideration of audio-visual asynchrony and the potential side effects of our proposed method. In **Appendix D, line 86**, an ablation study was conducted to alleviate this concern.
>
> We compare our final bidirectional approach with the unidirectional variants that only use either **audio-to-visual (A2V)** or **visual-to-audio (V2A)** spatial-channel-temporal attention mechanism. As indicated in **Table 3 (also depicted below)**:
>
> | Module |      | Segment-level |          |          |          |          | Event-level |          |          |          |          |
> | :----: | ---- | :-----------: | -------- | -------- | -------- | -------- | ----------- | -------- | -------- | -------- | -------- |
> |  A2V   | V2A  |       A       | V        | AV       | Type     | Event    | A           | V        | AV       | Type     | Event    |
> |   -    | -    |     57.8      | 56.3     | 49.8     | 55.2     | 54.9     | 48.2        | 51.7     | 43.9     | 48.8     | 47.6     |
> |   √    | -    |     56.4      | **59.5** | **53.3** | 56.4     | 55.0     | 47.4        | 55.9     | **46.3** | 49.9     | 47.8     |
> |   -    | √    |   **59.4**    | 57.3     | 50.8     | 55.8     | 56.8     | 49.2        | 54.1     | 44.4     | 49.2     | 48.6     |
> |   √    | √    |     59.0      | 59.4     | 52.8     | **57.1** | **57.0** | **49.2**    | **56.1** | 46.1     | **50.5** | **49.1** |
>
> we observe that using the A2V module alone does not significantly decrease the accuracy for visual and audio-visual events. However, without visual guidance (V2A), the performance of audio events suffers a considerable decline; Likewise, the performance of visual events drops without audio guidance (using the V2A module alone). **These experimental findings demonstrate the necessity of visual guidance for audio events and the need for audio guidance for visual events.** Our proposed DG-SCT model can bidirectionally guide the representation of each modality, thus enhancing the accuracy of downstream audio-visual tasks.
>
> Based on our analysis, we have determined that the benefits of incorporating bidirectional modality guidance for capturing rich information outweigh the interference caused by audio-visual asynchrony. However, we acknowledge the importance of addressing the issue of audio-visual asynchrony in future work. Specifically, we plan to explore techniques during the representation phase to mitigate this problem, aiming to further enhance the performance of our model in the future.
>
> Moreover, we have added experiment on **VGG-Sound (40K) classification** (VGG-Sound dataset is the subset of AudioSet). The results are as follows:
>
> | Method |   Acc    |
> | :----: | :------: |
> |  A+V   | **69.7** |
> |   A    |   63.8   |
> |   V    |   60.6   |
>
> The results demonstrate that incorporating both audio and visual information simultaneously leads to improved performance in audio-visual joint classification task.
>
> ------
>
> minor issue: **"Page 2, line 54, it should be HTS-AT, not HT-SAT."** We would like to extend our sincere appreciation for your dedicated effort in carefully reviewing our work! This issue has been addressed.

---

> > ### Comment · Reviewer_BzrV · 2023-08-19
> > **Discussion**
> >
> > Dear authors,
> >
> > Thanks so much for the explanation and for pointing me to the appendix.
> >
> > >In joint classification task, mutual information is critical as well. For example, when considering only the visual modality, a radio appears motionless. However, by taking into account the audio, one can determine whether the radio is producing sound. Similarly, when only listening to the audio, the sound of turning on and off a light may sound similar.
> >
> > Don't these examples show that modality-unique information (rather than modality-mutual information) is important for joint classification? As the radio sound is missing in the visual modality, and the turning on/off information is missing in the visual modality? Do the authors actually mean "mutual object"?
> >
> > >It seems you are referring to the consideration of audio-visual asynchrony and the potential side effects of our proposed method. In Appendix D, line 86, an ablation study was conducted to alleviate this concern.
> >
> > This ablation is very interesting. I like this experiment.
> >
> > >Moreover, we have added experiment on VGG-Sound (40K) classification (VGG-Sound dataset is the subset of AudioSet). The results are as follows ...
> >
> > Is it true that VGG-Sound is a subset of AudioSet? If so, can the authors provide a reference on this? The original VGG-Sound is 200k, how the 40k set is selected? Is this fair to use the 40k results to compare to other papers reporting on 200k?
> >
> > Besides the question regarding the dataset, I actually think it is quite easy for a model to have better av performance than a or v-only performance, but what I really want to know is that - comparing a model that explicitly focuses on modality-mutual information (what is proposed) and a model without such focus (e.g., say MBT from Google, or a variant of the proposed method that without such focus), which would be better? I understand the authors may not have time to do this experiment, but want to know the authors' opinion.
> >
> > Anyways, I don't believe joint classification is a fatal problem of this paper, as many a-v applications do require mutual information.

---

> > > ### Author Response · Authors · 2023-08-20
> > >
> > > Dear Reviewer BzrV,
> > >
> > > Thank you so much for your reply! Let us respond to your questions point by point.
> > >
> > > > Don't these examples show that modality-unique information (rather than modality-mutual information) is important for joint classification? As the radio sound is missing in the visual modality, and the turning on/off information is missing in the visual modality? Do the authors actually mean "mutual object"?
> > >
> > > We strongly agree with the reviewer’s viewpoint that modality-specific information is indeed crucial for audio-visual tasks. However, it is undeniable that modality-mutual information also plays an important role in the context of audio-visual understanding. It enables the alignment and integration of both modalities, allowing the model to have a better understanding of audio-visual tasks. We apologize for any confusion caused by the lack of clarity in the examples we provided, which led to the reviewer’s misunderstanding. **While it is true that one can determine if a radio is emitting sound solely based on the audio modality, what about more challenging tasks? For instance, consider the task of locating a sound-emitting object within a video.** In such cases, audio can guide the visual modality, with a focus on the regions representing the radio object in the video and providing information on when the radio emits sound and when it doesn’t. This operation corresponds to the A2V module in our approach. Similarly, the V2A module enriches the audio modality with information guided by the visual modality.
> > >
> > > > Is it true that VGG-Sound is a subset of AudioSet? If so, can the authors provide a reference on this? The original VGG-Sound is 200k, how the 40k set is selected? Is this fair to use the 40k results to compare to other papers reporting on 200k?
> > >
> > > The origin of VGG-Sound can be traced back to the paper **“VGGSOUND: A LARGE-SCALE AUDIO-VISUAL DATASET.”** The paper does not explicitly mention whether VGG-Sound is a subset of AudioSet. The description in the paper states that VGG-Sound is a large-scale dataset consisting of over 200k video clips covering 300 audio classes sourced from YouTube videos. From what we recall, these 10-second videos are derived from processing AudioSet. However, since the paper lacks specific information on this matter, it is indeed an oversight on our part. We sincerely appreciate you bringing this issue to our attention. In the future, we may consider cross-referencing the YouTube IDs of the videos in VGG-Sound with those in AudioSet to validate whether VGG-Sound is a subset of AudioSet. Unfortunately, due to time constraints, we regret that we cannot provide you with a definitive answer at this moment.
> > >
> > > Regarding VGG-Sound 40k, it is derived from the paper **“Contrastive Positive Sample Propagation along the Audio-Visual Event Line.”** The paper collects the VGGSound-AVEL100k dataset, whereas our **VGG-Sound 40k dataset is 40,000 finely annotated videos of the VGGSound-AVEL100k provided by the authors**. It would not be fair to compare the results of the 40k dataset with other papers that report on the 200k dataset because the latter covers **300** classes while the former only comprises **141** classes.
> > >
> > > > Besides the question regarding the dataset, I actually think it is quite easy for a model to have better av performance than a or v-only performance, but what I really want to know is that - comparing a model that explicitly focuses on modality-mutual information (what is proposed) and a model without such focus (e.g., say MBT from Google, or a variant of the proposed method that without such focus), which would be better?
> > >
> > > We apologize for any misunderstanding and for not conducting the experiments you were hoping to see. In our opinion, models like MBT that do not explicitly focus on modality-mutual information may achieve comparable or even better results in coarse-grained tasks such as AudioSet and VGGSound. As you mentioned, these tasks often require only a single modality, where the goal is to extract information that is not present in one modality but can be found in the other.
> > >
> > > However, in more challenging downstream tasks such as AVE, AVVP, AVS, and AVQA, which are also the focus of our work, **our proposed method is expected to perform better**. Just like the aforementioned example, these tasks often require the integration of information from both modalities to achieve successful results.
> > >
> > > ------
> > >
> > > Thank you again for raising the questions, and we hope our response has been helpful to you. Finally, we wish you all the best!

---

> > > > ### Comment · Reviewer_BzrV · 2023-08-21
> > > > **discussion**
> > > >
> > > > >Regarding VGG-Sound 40k, it is derived from the paper “Contrastive Positive Sample Propagation along the Audio-Visual Event Line.” The paper collects the VGGSound-AVEL100k dataset, whereas our VGG-Sound 40k dataset is 40,000 finely annotated videos of the VGGSound-AVEL100k provided by the authors. It would not be fair to compare the results of the 40k dataset with other papers that report on the 200k dataset because the latter covers 300 classes while the former only comprises 141 classes.
> > > >
> > > > I think VGGSound is not a subset of AudioSet, but that is fine. I think evaluate the model on any large scale dataset like AudioSet or VGGsound is OK. Testing on more dataset would be better, but not a mandotory thing. In the final version, I hope the authors can be more careful in describing the setting.
> > > >
> > > > >We strongly agree with the reviewer’s viewpoint that modality-specific information is indeed crucial for audio-visual tasks. However, it is undeniable that modality-mutual information also plays an important role in the context of audio-visual understanding. It enables the alignment and integration of both modalities, allowing the model to have a better understanding of audio-visual tasks. We apologize for any confusion caused by the lack of clarity in the examples we provided, which led to the reviewer’s misunderstanding. While it is true that one can determine if a radio is emitting sound solely based on the audio modality, what about more challenging tasks? For instance, consider the task of locating a sound-emitting object within a video. In such cases, audio can guide the visual modality, with a focus on the regions representing the radio object in the video and providing information on when the radio emits sound and when it doesn’t. This operation corresponds to the A2V module in our approach. Similarly, the V2A module enriches the audio modality with information guided by the visual modality.
> > > >
> > > > >We apologize for any misunderstanding and for not conducting the experiments you were hoping to see. In our opinion, models like MBT that do not explicitly focus on modality-mutual information may achieve comparable or even better results in coarse-grained tasks such as AudioSet and VGGSound. As you mentioned, these tasks often require only a single modality, where the goal is to extract information that is not present in one modality but can be found in the other.
> > > >
> > > > The samples and not outperforming MBT is fine. I just want to double check with the author to avoid misunderstanding. When a model is focusing on one aspect, natually it would loss on some others, totally understanable, and it is not a major flaw of the paper. I hope the author could clarify this in the next version of the paper.
> > > >
> > > > >However, in more challenging downstream tasks such as AVE, AVVP, AVS, and AVQA, which are also the focus of our work, our proposed method is expected to perform better. Just like the aforementioned example, these tasks often require the integration of information from both modalities to achieve successful results.
> > > >
> > > > I agree and view it is the key strength of the paper.
> > > >
> > > > I thank the authors for the discussion again, and I will keep my positive rating.

---

> > > > > ### Author Response · Authors · 2023-08-21
> > > > >
> > > > > Dear Reviewer BzrV,
> > > > >
> > > > > Thank you so much for checking our response. We are glad that you keep the original positive rating.

---

### Decision · Program_Chairs · 2023-09-21

**Decision:**

Accept (poster)

**Comment:**

Initially, the reviews were mixed. The reviewers appreciated the audiovisual prompting idea, demonstrated results, and paper presentation. However, there were concerns regarding the evaluation. In particular, reviewer myp2 had concerns regarding how the number of parameters were controlled in the experiments. The rebuttal assuaged most of the raised concerns. During the post-rebuttal discussion, reviewer myp2 had remaining concerns that the trainable and total parameters are larger than the baselines and noted to double check the total number of parameters. The AC agrees with these points, and thinks that this point can be addressed in the final paper. The AC is convinced of the positive merits of the paper as noted in the reviews. Please take into account all reviewer feedback in the camera-ready version.